# Integrative characterization of the near-minimal bacterium *Mesoplasma florum*

Dominick Matteau[1], Jean-Christophe Lachance[1], Frédéric Grenier[1], Samuel Gauthier[1], James M Daubenspeck[2], Kevin Dybvig[2], Daniel Garneau[1], Thomas F Knight[3], Pierre-Étienne Jacques[1] & Sébastien Rodrigue[1,*]

## Abstract

The near-minimal bacterium *Mesoplasma florum* is an interesting model for synthetic genomics and systems biology due to its small genome (~ 800 kb), fast growth rate, and lack of pathogenic potential. However, fundamental aspects of its biology remain largely unexplored. Here, we report a broad yet remarkably detailed characterization of *M. florum* by combining a wide variety of experimental approaches. We investigated several physical and physiological parameters of this bacterium, including cell size, growth kinetics, and biomass composition of the cell. We also performed the first genome-wide analysis of its transcriptome and proteome, notably revealing a conserved promoter motif, the organization of transcription units, and the transcription and protein expression levels of all protein-coding sequences. We converted gene transcription and expression levels into absolute molecular abundances using biomass quantification results, generating an unprecedented view of the *M. florum* cellular composition and functions. These characterization efforts provide a strong experimental foundation for the development of a genome-scale model for *M. florum* and will guide future genome engineering endeavors in this simple organism.

**Keywords** *Mesoplasma florum*; Mollicutes; synthetic genomics; systems biology; whole-cell characterization

**Subject Category** Microbiology, Virology & Host Pathogen Interaction

**Mol Syst Biol. (2020) 16: e9844**

## Introduction

Since the first report of the *in vitro* synthesis of a complete gene (Agarwal *et al*, 1970), DNA synthesis and assembly techniques have improved considerably in terms of efficiency and capacity (Hughes & Ellington, 2017; Schindler *et al*, 2018). Large DNA molecules such as entire chromosomes can now be synthetized at reasonable cost,

enabling the creation of synthetic or semi-synthetic organisms, an emerging field known as synthetic genomics (Montague *et al*, 2012; van der Sloot & Tyers, 2017; Mitchell & Ellis, 2017; Schindler *et al*, 2018). Given proper design, synthetic organisms could play a very important role in addressing some of the most critical challenges of the 21st century such as the development of sustainable energy sources, the fight against antibiotic resistance, and the treatment of diseases such as cancer and diabetes (Khalil & Collins, 2010; Alper *et al*, 2010; Cambray *et al*, 2011).

Although the tools to build artificial chromosomes are now available, not even a handful of significantly modified synthetic genomes have been reported (Hutchison *et al*, 2016; Richardson *et al*, 2017; Fredens *et al*, 2019), and our ability to design complete genomes from scratch is extremely poor at best. Consequently, little is still truly understood about genome design principles. This is mainly explained by the overwhelming complexity of common model organisms, which outstrips our current analytical skills and inhibits our ability to rationally evaluate genome designs. Moreover, the number of possible artificial genome configurations can quickly become overwhelming, even for small genome bacteria. In that context, systems biology approaches such as genome-scale metabolic models (GEMs) could soon become powerful tools to systematically evaluate genome designs and help select the most promising scenarios for total synthesis (preprint: Chalkley *et al*, 2019; Rees-Garbutt *et al*, 2020). GEMs consist of mathematically structured knowledge frameworks describing the metabolism of organisms, offering phenotypic predictions capabilities useful in a wide-range of applications from omics data integration to metabolic engineering (Oberhardt *et al*, 2009; Durot *et al*, 2009; Bordbar *et al*, 2014; O'Brien *et al*, 2015; Ebrahim *et al*, 2016; Kim *et al*, 2016; Gu *et al*, 2019). For example, the impact of multiple gene deletions or environmental stresses on metabolic fluxes and growth rate can be predicted, providing context-specific hypotheses prior to experimental testing. To perform accurate predictions, GEMs must however be constrained and validated by experimental data such as the biomass composition of the cell (% of DNA, RNA, proteins, etc.) (Feist & Palsson, 2010; Lachance *et al*, 2019b). To date, more than 100 high-quality GEMs have been reconstructed, including GEMs for many

---

1  Département de biologie, Université de Sherbrooke, Sherbrooke, QC, Canada
2  Department of Genetics, University of Alabama at Birmingham, Birmingham, AL, USA
3  Ginkgo Bioworks, Boston, MA, USA
   *Corresponding author. Tel: +1 819 821 8000 ext 62939; Fax: +1 819 821 8049; E-mail: sebastien.rodrigue@usherbrooke.ca

model organisms such as *Escherichia coli*, *Saccharomyces cerevisiae*, and *Homo sapiens* (Norsigian *et al*, 2020). GEMs have also been extended to include additional cellular processes such as proteome expression, thereby increasing their capabilities and breadth of applications (King *et al*, 2015; O'Brien & Palsson, 2015).

Because of their exceptionally small genomes (0.58–2.2 Mbp) (Sirand-Pugnet *et al*, 2007), near-minimal bacteria of the Mollicutes class have long been proposed as models to study the basic principles of life (Morowitz, 1984). These very small (0.2–0.6 μm) wall-less bacteria do not constitute ancient or primitive forms of life but rather evolved from low G-C content Gram positive bacteria through a process of massive gene loss (Pettersson & Johansson, 2002; Maniloff, 2002). This resulted in a drastic simplification of their metabolism, with many incomplete or missing metabolic pathways (Dybvig & Voelker, 1996; Pollack *et al*, 1997). The genomic simplicity of Mollicutes thus offers a unique opportunity to achieve an unprecedented characterization of cellular processes, reduces the number of artificial genome configurations to be tested using synthetic genomics approaches, and decreases the costs related to chromosome synthesis (Xavier *et al*, 2014; Lachance *et al*, 2019a). Among all Mollicutes, members of the *Mycoplasma* genus are the most extensively studied, with many species infecting various animals, including humans (Dybvig & Voelker, 1996; Maniloff, 2002). However, mycoplasmas recently gained particular attention with the development of whole-genome chemical synthesis, assembly, and cloning in yeast (Gibson *et al*, 2008; Gibson & Benders, 2008; Benders *et al*, 2010). The total synthesis and cloning of the 1.08 Mb *Mycoplasma mycoides* subspecies *capri* GM12-based genome followed by its transplantation into a recipient bacterium (*Mycoplasma capricolum* subspecies *capricolum*) notably led to the creation of the first cell controlled by an entirely synthetic chromosome, JCVI-syn1.0 (Gibson *et al*, 2010; Sleator, 2010). This impressive *tour de force* recently culminated with the creation of the first artificial "working approximation" of a minimal cell, JCVI-syn3.0 (Hutchison *et al*, 2016). This minimal bacterium harbors a reduced and synthetic version of the *M. mycoides* subspecies *capri* genome totalizing 531 kb and 473 genes (GenBank: CP014940.1), making it the smallest genome ever observed in any autonomously replicating cell (Hutchison *et al*, 2016; Glass *et al*, 2017). The JCVI-syn3.0 strain however showed altered morphological traits and impaired growth rates compared with the *M. mycoides* parent strain (doubling time of ~ 2–3 h vs. ~ 1 h), which were restored by the incorporation of 19 additional genes (Breuer *et al*, 2019). The resulting strain, named JCVI-syn3A, carried a genome of 543 kb and 493 genes (GenBank: CP016816.2).

First described in 1984 as *Acholeplasma florum* (McCoy *et al*, 1984), the near-minimal bacterium *Mesoplasma florum* constitutes another member of the Mollicutes class particularly well suited for synthetic genomics and systems biology studies. While closely related to *M. mycoides*, *M. florum* however has a smaller genome, shows faster growth rates, and has no pathogenic potential (Sirand-Pugnet *et al*, 2007; Gibson *et al*, 2010; Matteau *et al*, 2015; Baby *et al*, 2018). The genome of the L1 type strain, for example, comprises only 793 kb and 720 predicted genes (GenBank: AE017263.1), while the genome of the *M. mycoides capri* LC GM12 accounts for 1.09 Mb and 879 genes (GenBank: CP001621.1). These features greatly facilitate the manipulation of *M. florum* and its distribution throughout the scientific community. As for most

Mollicutes, *M. florum* also uses an alternative genetic code (*Mycoplasma/Spiroplasma* code) that limits the exchange of genetic material from and to other microorganisms (Navas-Castillo *et al*, 1992). Importantly, genetic manipulation tools have recently been developed specifically for this bacterium, including procedures for whole-genome cloning in yeast and genome transplantation (Matteau *et al*, 2017; Baby *et al*, 2017). Gene conservation and essentiality analyses have also showed that 57 putatively essential *M. florum* genes have no homolog in the synthetic JCVI-syn3.0 strain, suggesting that different minimal genome compositions and configurations probably exist, even within closely related species (Baby *et al*, 2018). In addition, these analyses enabled the formulation of different genome reduction scenarios for *M. florum*, providing starting points for genome minimization efforts (Baby *et al*, 2018). The comparison of the JCVI-syn3.0 genome with other minimal genomes offers a unique opportunity to decipher genome design principles and some of the most fundamental principles of life, in the same way that the two first complete bacterial genome sequences provided insights about the minimal gene set required for cellular life (Mushegian & Koonin, 1996).

Here, we report the first integrative characterization of *M. florum* to advance fundamental knowledge on this emerging model. More specifically, we accurately measured several physical and physiological parameters of *M. florum* L1 growing in rich medium, including the cell diameter, buoyant density, dry mass, optimum growth temperature, growth rate, and growth kinetics. We also defined the macromolecular composition of the cell, identified and characterized more than 400 active promoters, and proceeded to the reconstruction of *M. florum* transcription units (TUs). Finally, we used transcriptomics and proteomics expression datasets to estimate RNA and protein species abundances, revealing the relative importance of the different cellular processes of a near-minimal cell. Our work contributes to a detailed understanding of global cell functioning in a simple organism and provides an experimental foundation for the development of a systems and synthetic biology platform.

## Results

### *Mesoplasma florum* optimal growth temperature and growth kinetics

The doubling time and the optimal growth temperature represent fundamental parameters in the characterization of a bacterial strain. Moreover, the doubling time is a critical constraint in many cellular modeling approaches such as GEMs (Feist & Palsson, 2010; Lachance *et al*, 2019b). Accurate measurement of these parameters has however never been reported specifically for *M. florum*. The optimal growth temperature of the type strain *M. florum* L1 was therefore evaluated in ATCC 1161 medium by measuring the doubling time at different incubation temperatures typically used for Mollicutes (~ 30–38°C) (Brown *et al*, 2007). Doubling times were determined using colorimetric assays that measure the time needed for twofold culture dilutions to reach the same optical density at 560 nm ($OD_{560\ nm}$). Raw *M. florum* growth curves are presented in Appendix Fig S1. The smallest doubling time was observed at a temperature of 34°C (38 ± 5 min) while no growth was observed at a temperature higher than 36°C (Fig 1A), contrasting with

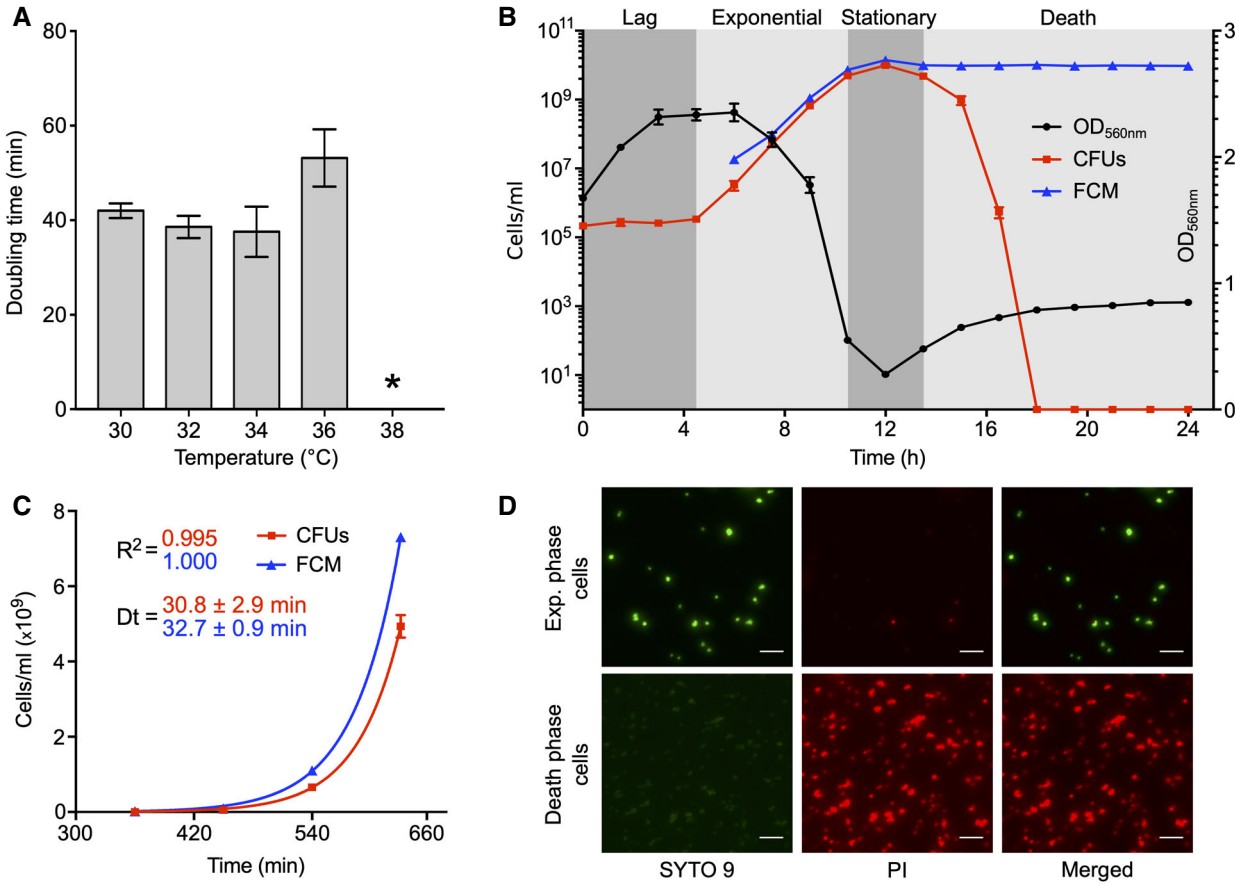

**Figure 1. Analysis of *M. florum* growth in ATCC 1161 medium.**

A *M. florum* doubling time at different incubation temperatures measured by colorimetric assays. The bars represent the mean and standard deviation values obtained from three technical replicates. The asterisk indicates the absence of significant growth, preventing the calculation of a doubling time.

B *M. florum* growth kinetics at 34°C. Growth was monitored for 24 h by measuring the optical density at 560 nm (black circles) as well as the cell concentrations using two different methods, colony-forming units (CFUs, red squares) and flow cytometer (FCM, blue triangles). The four typical bacterial growth phases (lag, exponential, stationary, and death) are represented by gray shading. The dots and error bars indicate the mean and standard deviation values obtained from three independent biological replicates. CFU data points superimposed to the $x$-axis represent values below the limit of detection ($2 \times 10^{-2}$).

C Exponential growth fit on CFU (red squares) and FCM (blue triangles) counts shown in B. Calculated doubling times (Dt) and correlation coefficients ($R^2$) are shown. The dots and error bars indicate the mean and standard deviation values obtained from three independent biological replicates.

D Representative images of SYTO 9 and propidium iodide (PI) double-stained *M. florum* cells, harvested from an exponential or death-phase culture, observed by widefield fluorescence microscopy. The brightness of each channel was adjusted equally between conditions. Scale bar: 5 μm.

pathogenic mycoplasmas such as *M. mycoides*, *M. capricolum*, and *Mycoplasma pneumoniae*. These results are consistent with previous observations concerning different *M. florum* strains (McCoy *et al*, 1984) and other members of the *Mesoplasma* genus (Tully *et al*, 1994).

We then used flow cytometry (FCM) and colony-forming units (CFUs) to precisely measure the growth kinetics of *M. florum* incubated at the optimal growth temperature (34°C). We first validated that cell concentrations measured by FCM were well correlated with culture dilutions (Appendix Fig S2). By following cell concentrations over ~ 24 h, we could observe an overall pattern corresponding to the four typical bacterial growth phases (Fig 1B). The exponential phase coincided with a substantial drop in medium pH (from ~ 8.0 to 6.5) causing the phenol red present in the culture medium to change color from red to orange, corresponding to an important decrease in measured $OD_{560\ nm}$. Using exponential curve fitting on

FCM and CFUs data, we determined a doubling time of 30.8 ± 2.9 min and 32.7 ± 0.9 min, respectively (Fig 1C). CFU and FCM cell concentrations were highly consistent with each other until late stationary phase, where they both reached a plateau at ~ 1 × $10^{10}$ cells/ml and started to diverge. The stationary phase was also marked by the lowest $OD_{560\ nm}$ value observed for the entire experiment, corresponding to a yellow medium color and a medium pH around 6.0. This was followed by a gradual formation of cell aggregates in the culture, resulting in a notable increase in the measured $OD_{560\ nm}$. This phenomenon was accompanied by a rapid diminution of CFU counts, suggesting an important loss in cell viability reminiscent of the death phase (Fig 1B). We validated that the decrease in CFU counts was effectively due to an altered cell viability by SYTO 9 and propidium iodide (PI) dual staining fluorescence microscopy (Fig 1D). As expected, *M. florum* cells harvested at the death phase showed an intense PI signal and practically no SYTO 9

fluorescence, indicating a significantly compromised cell membrane integrity. Similar signals were observed for formaldehyde fixed and permeabilized cells (Appendix Fig S3), whereas exponential-phase cells showed a strong SYTO 9 fluorescence and almost no PI signal, typical of healthy cells (Fig 1D).

## Physical characteristics and macromolecular composition of the cell

To better define the physical constraints shaping the biology of *M. florum*, we sought to precisely measure its cell diameter since the only quantitative data available for this species relied on filtration studies (McCoy *et al*, 1984). Filtration constitutes an indirect approach that can be subjected to different sources of variation such as pore size heterogeneity and deformation of cellular morphology, especially for wall-less bacteria. We analyzed exponential-phase *M. florum* cells using two different techniques, transmission electron microscopy (TEM) and stimulated emission depletion (STED) microscopy. Cells were stained with PicoGreen and mCLING (Revelo *et al*, 2014), respectively, targeting the DNA and the cellular membrane, prior to STED microscopy examination. Representative images obtained from both techniques are shown in Fig 2A and B. Both TEM and STED microscopy showed predominantly ovoid cells, with a cell diameter ranging from approximately 300 to 600 nm and 500 to 1,000 nm, respectively (Fig 2C). An average cell diameter of $434 \pm 53$ nm was observed for TEM and $741 \pm 98$ nm for STED microscopy (Fig 2D). The significant difference observed by the two methods is most likely caused by biases associated with sample preparation. TEM, for example, requires a dehydration of the cells with ethanol, which can cause cell shrinkage and therefore a reduction in their apparent diameter (Zhang *et al*, 2017). STED, on the other hand, requires the use of a mounting media during slide preparation that can cause sample distortion and alteration of morphological features (Peterson *et al*, 2015; Fouquet *et al*, 2015; GE, 2018). Interestingly, TEM pictures also showed evidences of a polysaccharidic layer on the periphery of *M. florum* cells, a morphological feature shared by many Mollicutes including the closely related *M. mycoides* and *M. capricolum* (Bertin *et al*, 2013; Gaurivaud *et al*, 2014; Daubenspeck *et al*, 2014; Bertin *et al*, 2015).

Measuring the total mass of a cell requires specialized equipment and can be very challenging, especially for small cells (Bryan *et al*, 2014; Zhao *et al*, 2014; Rahman *et al*, 2015). The cell mass can however be estimated using different mathematical equations that involve only a limited number of variables more easily amenable to quantification, including the cell diameter, buoyant density, and dry mass. Since we had already measured the cell diameter of *M. florum* using TEM and STED microscopy, we next evaluated its buoyant density by discontinuous Percoll density gradient centrifugation. After one or two rounds of centrifugation, the *M. florum* cell pellet was located at the bottom of the 1.05 g/ml Percoll layer, indicating a buoyant cell density lying between 1.05 and 1.08 g/ml (Fig 2E and Table 1). We next determined the *M. florum* cell dry mass using conventional weighting procedures performed on exponential-phase batch cultures (see Materials and Methods and Fig EV1), and observed a total dry mass of $22.1 \pm 4.2$ fg per cell (Fig 2F and Table 1). The measured buoyant cell density and cell dry mass were then used to infer the most probable *M. florum* cell mass using four different equations (see Equations 4–7 in Materials and Methods section). Three of those equations also require the total dry mass fraction and the dry mass density to estimate the total mass of the cell, which were assumed to be within typical ranges found in bacteria, i.e., 20–30% and 1.3–1.5 g/ml, respectively (Bakken & Olsen, 1983; Bratbak & Dundas, 1984; Bratbak, 1985; Fischer *et al*, 2009; Bionumbers, 2015). Interestingly, all four equations converged to a relatively tight range of cellular mass (88.2–103.3 fg), which corresponded to a cell diameter (538–570 nm) positioned in-between average values obtained by TEM and STED microscopy and within the overlapping portion of their relative distribution (Fig 2C and G, and Table 1). Refining the cell diameter also allowed the estimation of the most probable cell volume (0.082–0.097 µm³), cell surface area (0.911–1.021 µm²), and surface area to volume ratio (*SA:V*; 10.5–11.1 µm⁻¹) using Equations 1–3 (see Materials and Methods), respectively (Table 1).

---

**Figure 2. *M. florum* physical characteristics.**

A Representative image of *M. florum* cells observed by transmission electronic microscopy (TEM). Scale: 100 nm.

B Representative image of PicoGreen (DNA) and mCLING (cellular membrane) double-stained *M. florum* cells observed by stimulated emission depletion (STED) microscopy. Scale: 1 µm.

C Frequency distribution of *M. florum* average cell diameter measured by TEM and STED as shown in A and B, respectively. The average cell diameter was obtained by averaging the minor and major axis values measured for each cell. A Gaussian curve fit is indicated for each method, and the calculated correlation coefficients are shown. Bins: 50 nm.

D Boxplots showing the median and interquartile range of the average cell diameter calculated from 389 and 169 individual cells analyzed by TEM and STED, respectively. Whiskers indicate the 10–90 percentile range.

E Picture of *M. florum* cells analyzed by discontinuous density gradient centrifugation in Percoll. The approximative density of each Percoll layer is indicated (g/ml) and colored in blue if trypan blue was added to the layer. The position of the cell pellet is marked.

F *M. florum* biomass quantification. The mass of each macromolecular constituent is shown as well as its relative fraction in the quantified cellular dry mass. Bars represent the mean and standard deviation values obtained from three independent biological replicates (dry mass) or four technical replicates (proteins, RNA, lipids, DNA, carbs). The "Other" category bar represents the residual mass obtained by the subtraction of all quantified macromolecule masses from the total dry mass value.

G Graph showing the relation between the *M. florum* cell diameter (*d*) and its cell mass (*CM*) according to cell mass Equations 4–7 (see Materials and Methods). For each equation, the mean cell mass (*CM_mean*) is indicated by a colored line, and the range of probable values (*CM_min − CM_max*) is shown by a light-colored shading. The mean values of the average cell diameter measured by TEM and STED (see panel D) are indicated by black dashed lines. The portion of the graph where all the *CM_mean* curves converge is enlarged and devoid of colored shadings for representation purposes. *CM_mean* interception points encompassing all other interception points are encircled, and their corresponding *x* and *y* coordinates are indicated by fuchsia dashed lines (most probable cell diameter and most probable cell mass ranges).

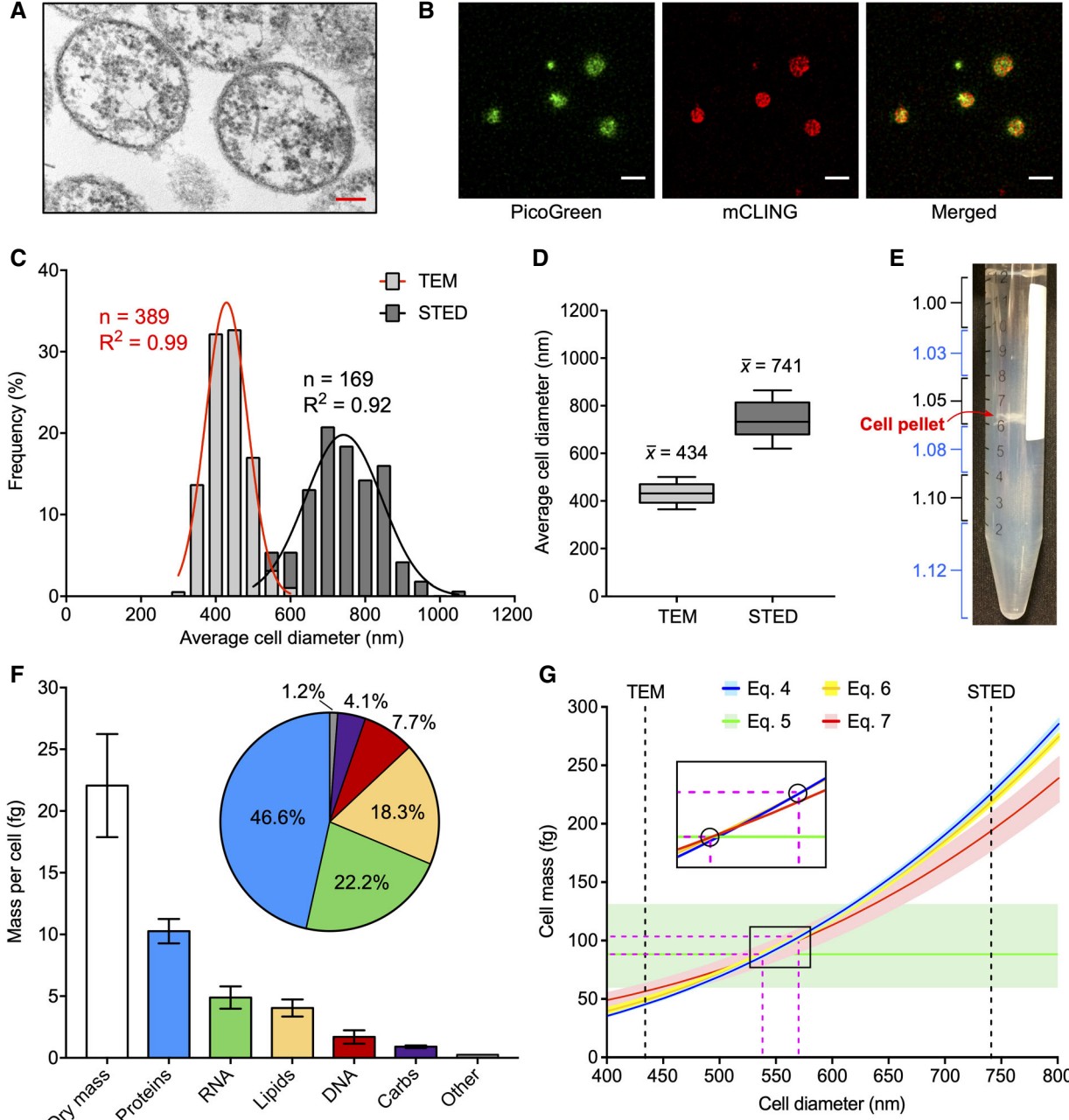

**Figure 2.**

The vast majority of the cell dry mass can be divided into four classes of macromolecules: proteins, lipids, nucleic acids, and carbohydrates (Cooper & Hausman, 2000). To better define the *M. florum* dry mass, we quantified each of these macromolecules using different high sensitivity quantification assays and gas chromatography-mass spectrometry (GC-MS) methods (see Materials and Methods and Fig EV1). According to our analysis, nearly two-thirds of the total dry mass was composed of proteins and RNA, with a relative abundance of approximately 46.6 and 22.2%, respectively (Fig 2F and Table 1). The remaining fraction of the dry mass was divided as follows: 18.3% for lipids (Dataset EV8), 7.7% for DNA, and 4.1% for carbohydrates. Overall, these results are

comparable to fractions observed in other Mollicutes species (Razin *et al*, 1963). The majority of the *M. florum* carbohydrate fraction most probably accounts for the polysaccharidic layer observed by TEM. Carbohydrates detected by mass spectrometry were mainly composed of galactose (0.50 ± 0.07 fg), glucose (0.19 ± 0.03 fg), rhamnose (0.18 ± 0.01 fg), and mannose (0.04 ± 0.01 fg), representing approximately 54.9, 20.6, 20.0, and 4.5% of the total carbohydrate mass, respectively. Interestingly, the residual dry mass, i.e., the difference between the quantified dry mass and the sum of all quantified macromolecules, represented only 1.2% (0.26 fg) of the total dry mass, most likely accounting for small molecules, metabolites, cofactors, and ions (Fig 2F).

Table 1. Summary of *Mesoplasma florum* biomass composition and physical characteristics measured or estimated in this study.

| Cellular biomass | Mean ± SD (fg) | Physical parameters | Most probable values |
|---|---|---|---|
| Dry mass | 22.1 ± 4.2 | Density | 1.05–1.08 g/ml[a] |
| Proteins | 10.3 ± 1.0 | Cell diameter | 538–570 nm[b] |
| RNA | 4.9 ± 0.9 | Cell mass | 88.2–103.3 fg[b] |
| Lipids | 4.0 ± 0.7 | Cell volume | 0.082–0.097 μm³ [c] |
| DNA | 1.7 ± 0.5 | Cell surface area | 0.911–1.021 μm² [c] |
| Carbohydrates | 0.9 ± 0.1 | *SA:V* | 10.5–11.1 μm⁻¹ [c] |

[a]Measured by discontinuous Percoll density gradient centrifugation.
[b]Estimated using cell mass Equations 4–7 (see Fig 2G and Materials and Methods).
[c]Inferred from the most probable cell diameter using Equations 1–3 (see Materials and Methods).

## Genome-wide identification of promoters

Transposon mutagenesis and gene conservation datasets have recently been published for *M. florum* and allowed the proposition of different genome reduction scenarios for this bacterium (Baby *et al*, 2018). However, these predictions did not account for promoter organization and therefore retained all intergenic regions in the reduced genome designs. The identification of all *M. florum* promoters and corresponding transcription units (TUs) would certainly improve the quality and accuracy of these predictions, in addition to providing highly valuable information about the transcriptome of this near-minimal cell. We therefore proceeded to the cartography of all *M. florum* transcription start sites (TSSs) at single nucleotide resolution using a previously described genome-wide 5′-rapid amplification of cDNA ends (5′-RACE) method (Carraro *et al*, 2014; Matteau & Rodrigue, 2015). Following Illumina sequencing (see Appendix Table S1 for a summary of library statistics), the number of read starts per million of mapped reads (RSPM) was calculated for each genomic position in a strand-specific manner, resulting in a frequency distribution reminiscent of a Poisson distribution (Appendix Fig S4A). Out of 1,586,448 possible sites (genome size multiplied by two to account for both strands), a total of 68,650 sites had a non-null TSS signal, of which 1,514 (< 0.1% of all sites) displayed a significant intensity (see Appendix Fig S4B and Materials and Methods for further details). This resulted in the identification of 605 candidate TSSs distributed throughout the *M. florum* chromosome (Fig 3A). DNA sequence analysis using the MEME software (Bailey & Elkan, 1994) revealed a conserved promoter motif present in 422 candidate TSSs highly reminiscent of promoter sequences identified in other Mollicutes species (Fig 3B and C), including *M. pneumoniae*, *Mycoplasma hyopneumoniae*, *Acholeplasma laidlawii*, and *Mycoplasma gallisepticum* (Weiner III, 2000; Güell *et al*, 2009; Weber *et al*, 2012; Yus *et al*, 2012; Mazin *et al*, 2014; Lloréns-Rico *et al*, 2015; Fisunov *et al*, 2016). More precisely, this promoter motif contained a −10 box typical of the sequences recognized by the principal σ factor in most bacteria (TAWAAT) (Helmann, 1995; Shultzaberger *et al*, 2007), as well as a partially degenerated TGN extension of the −10 box (EXT element) (Fig 3B). No clear evidence of a conserved −35 box emerged from the analysis. The occurrence of this promoter

sequence was validated in ~ 85% (357) of cases using the MAST software (Bailey & Gribskov, 1998), which also provided evidences for an additional 10 sites not initially included in the MEME constructed motif, for a grand total of 432 motif-associated TSSs (Fig 3C and Dataset EV1). No promoter motif could be identified for the remaining TSS candidates, suggesting a higher sequence variability at these sites or experimental artifacts.

As expected, the vast majority (78.0%) of motif-associated TSSs were located within intergenic regions of the chromosome (gTSSs), even though these regions occupy only ~ 6.1% of the genome (Fig 3D) (Baby *et al*, 2018). Interestingly, putative TSSs devoid of a promoter motif were located within coding sequences (CDS) in more than 90% of all instances, clearly contrasting with motif-associated TSSs (Fig EV2A). In most cases (76.2%), motif-associated gTSSs were in the same orientation (parallel) as their closest downstream gene (p-gTSS), with only a few cases (1.9%) of antiparallel downstream associated gene (a-gTSS) (Fig 3D). The remaining TSSs (22.0%) were found to be internal to coding regions of the genome (iTSS), most of the time in the same orientation as the overlapping gene in which they occur (p-iTSS). In total, nearly 12% of *M. florum* genes contained at least one motif-associated iTSS (Fig EV3D). p-iTSSs were found to be remarkably enriched near the end of their overlapping gene (Fig EV4C), with several instances separated by less than 100 bp from the next correctly oriented downstream gene (see Fig EV4D for a visual example). A few cases of p-iTSS were also precisely located on the first base of translation start codons, suggesting the transcription of leaderless mRNA (Weiner III, 2000; Moll *et al*, 2002; Zheng *et al*, 2011; Nakagawa *et al*, 2017) (Fig EV4C and E). gTSSs and iTSSs shared approximately the same distribution regarding their relative spacing with the conserved promoter motif, predominantly separated by 6 or 7 bases from the −10 box most proximal extremity (Fig 3E). Both TSS types were also located preferentially on coordinates corresponding to purine nucleotides (A or G), yet with an important bias for adenine (~ 70% of cases), reflecting the low G-C nature of the *M. florum* genome (Fig 3F). Despite these similarities, motif-associated gTSSs displayed a significantly higher signal intensity compared with motif-associated iTSSs, the latter group being not significantly different from TSSs without promoter motif (Fig EV2B). TSSs lacking the *M. florum* promoter motif were however not enriched for purine nucleotides like motif-associated gTSSs and iTSSs (Fig EV2C). Further information about the genetic context of gTSSs and iTSSs can be found in Appendix Supplementary Text and in Figs EV3 and EV4.

To validate promoters identified by 5′-RACE, we performed directional RNA sequencing (RNA-seq) on three exponential-phase *M. florum* steady-state cultures and evaluated read coverage across the genome. RNA-seq libraries were prepared in duplicate for each biological replicate, resulting in a total of six replicates. A statistical summary of RNA-seq libraries is presented in Appendix Table S1. We observed excellent correlations between the read coverage of the different replicates calculated on non-overlapping 1 kb windows (average Pearson correlation of 0.92), indicating a very good reproducibility of the method (Appendix Fig S5A). More importantly, coordinates of motif-associated TSSs coincided with a sharp increase in RNA-seq signal intensity calculated over the merged replicates, corroborating 5′-RACE identification results (Fig 3G). This feature was also observed for gTSSs and iTSSs analyzed independently, but to a much lesser extent in the case of iTSSs because

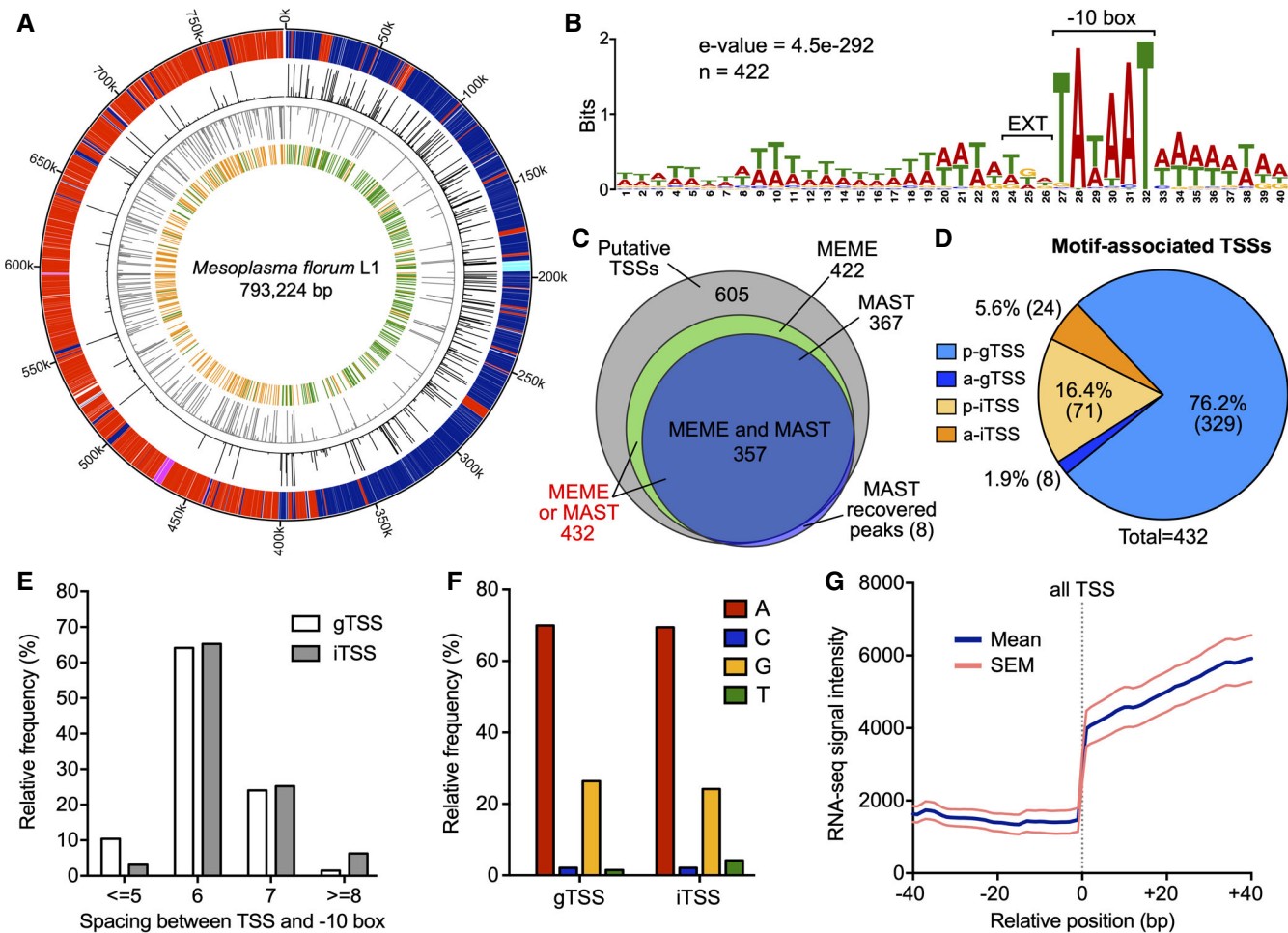

**Figure 3. Identification and analysis of *M. florum* promoters.**

A   Circular representation of the *M. florum* L1 chromosome enhanced with 5′-RACE data generated in this study. Outer to inner circle: genomic coordinates (kbp); genes encoded on the positive (blue for coding sequences, turquoise for RNAs) and negative (red for coding sequences, fuchsia for RNAs) DNA strands; raw 5′-RACE signal (0–1,000 read starts scale) observed at each genomic position for the positive (black) and negative (gray) DNA strands; putative transcription start sites (TSSs) identified on the positive (green) and negative (orange) DNA strands from significant 5′-RACE peaks.

B   *M. florum* promoter motif determined using the MEME software (Bailey & Elkan, 1994) from the DNA sequence located upstream the 605 putative TSSs identified by 5′-RACE. A total of 422 sites across the genome were included in the motif. The position of the −10 box (TAWAAT) and the extended element (EXT) is indicated.

C   Venn diagram illustrating the number of TSSs associated with a conserved promoter motif (see panel B) found by MEME, MAST, or both software compared with the total number of putative TSSs passing filters (605). Eight additional putative TSSs were added to the initial set according to the MAST search.

D   Localization and orientation of TSSs associated with a MEME or MAST promoter motif. p-gTSS, parallel intergenic TSS; a-gTSS, antiparallel intergenic TSS; p-iTSS, parallel internal TSS; a-iTSS, antiparallel internal TSS. For gTSSs, the orientation was defined according to the closest downstream gene, while the overlapping gene was used in the case of iTSSs.

E   Relative frequency distribution of the spacing between TSSs and their associated promoter −10 box.

F   Nucleotide identity at the transcription initiation site (+1) for gTSSs and iTSSs associated with a promoter motif.

G   Aggregate profile showing the mean RNA-seq read coverage observed at and around all motif-associated TSSs identified in this study. The calculated SEM is also shown. The aggregate profile was centered on the TSSs coordinates (relative position 0 bp), indicated by a gray dashed line.

of their intragenic context (Appendix Fig S6). Taken together, these results showed that motif-associated iTSSs and gTSSs share similar features and could both be responsible for the transcription of downstream genes.

## Reconstruction of transcription units

Having identified the key features of the *M. florum* promoters as well as the genomic coordinates of TSSs, we leveraged this

information to reconstruct TUs of this quasi-minimal bacterium. A TU consists of a DNA segment transcribed into a single mRNA molecule from one promoter to a transcription termination site (TTS) and encoding for zero, one or many open reading frames (ORFs). In Mollicutes, termination of transcription is believed to occur through a Rho-independent mechanism since no Rho protein homologue is detected in their genomes (de Hoon *et al*, 2005; D'Heygère *et al*, 2013). This mechanism involves structured terminators that can be reliably predicted from the DNA sequence and genes annotation,

reaching excellent sensitivity for many species such as *M. florum* (de Hoon *et al*, 2005). We therefore used an updated version of an algorithm developed by de Hoon and colleagues to predict the position of terminators in *M. florum* according to our previously published genome annotation (de Hoon *et al*, 2005; Baby *et al*, 2018). In total, 298 different Rho-independent terminators were predicted for the entire genome (Dataset EV2). As expected, the positions of the predicted terminators concurred with an important decrease in the RNA-seq signal intensity, supporting the predictions made by the algorithm (Appendix Fig S7). We then used the 432 motif-associated TSSs (gTSSs and iTSSs) identified by 5′-RACE along with the predicted transcription terminators to reconstruct all possible TUs (see Appendix Fig S8 and Materials and Methods for a detailed description of the procedure). After manual curation, a total of 387 TUs, each responsible for the expression of at least one gene, were reconstructed (Dataset EV3). These TUs encompassed more than 90% of all annotated *M. florum* genes (652), including all rRNA and tRNA genes, leaving only 68 genes out of 720 without an associated promoter (orphan gene). TUs start and stop coordinates coincided with a steep increase and decrease in the average RNA-seq read coverage (Fig 4A). Almost half of reconstructed TUs contained only a single gene, with up to 21 genes transcribed within a single RNA molecule, for an average of approximately 2.2 genes per TU (Fig 4B). The size of gene-associated TUs ranged from 112 bp to 12.5 kb and showed an average length of ~ 2.4 kb (Fig 4C) with 5′ and 3′ untranslated regions (UTR) of 58 and 51 bp, respectively (Fig 4D). Representative *M. florum* TUs are depicted in Fig 4E along with the associated 5′-RACE, terminators, and RNA-seq data.

As expected, most gene-encoding TUs were transcribed from gTSSs (86.6%) since they constitute the majority of TSSs identified in *M. florum* (Figs 3D and 4F). The remaining TUs were associated with p-iTSS (12.9%) and a-iTSS (0.5%). Both gTSS and iTSS-driven TUs showed enrichment of RNA-seq coverage, yet with a less defined 5′ border for iTSS TUs (Appendix Fig S9). A small number of mapped TSSs (56), mostly iTSSs (45 out of 56), could not be attributed to any downstream gene according to their genetic context. These TSSs were either (i) located within an intergenic region immediately upstream a predicted terminator; (ii) located within a gene positioned at the end of a TU; or (iii) facing a gene in the opposite direction. The two first cases were categorized as non-coding TUs, whereas TSSs facing a gene in the opposite direction were classified as orphan TSSs (Dataset EV4). Nonetheless, orphan TSSs and gTSSs located immediately before a terminator coincided with a small (~ 50–75 bp) RNA-seq signal enrichment (Appendix Fig S10). Some of these TSSs could be responsible for the expression of small non-coding RNAs (sRNAs) or antisense RNAs (asRNAs). Of the 652 genes covered by TUs, nearly two-thirds were individually included in only one TU, i.e., being transcribed from a single promoter (Fig 4G). The remaining genes were found to be comprised in up to four different TUs each. Interestingly, the vast majority of genes associated with an iTSS were also found to be transcribed from a gTSS, revealing only 15 genes exclusively transcribed from iTSSs (Figs 4H and EV4E). In fact, every gene associated with more than one TUs was part of a gTSS TU, and only about half of them (45.4%) were also transcribed from an iTSS TU. Overall, this suggests that iTSSs might have only a secondary role in the transcription of downstream genes. Nevertheless, iTSSs could still be involved in the transcription of other elements such as sRNAs.

## Estimation of intracellular levels of protein and nucleic acid species

We then estimated the intracellular levels of *M. florum* nucleic acid and protein species using our macromolecular biomass quantification data, starting with the DNA fraction. In *M. florum* L1, the genome is organized as a single and circular chromosome of 793,224 bp (Baby *et al*, 2013, 2018). Based on its sequence, this chromosome has a predicted molecular weight of 489,954 kDa. The number of chromosome copies can then be directly estimated from the DNA mass per cell in respect with its molecular weight. Given that *M. florum* contains $1.70 \pm 0.54$ fg of DNA per cell during the exponential phase (Table 1), we estimated that the average *M. florum* cell should contain the equivalent of 2.1 chromosome copies under these growth conditions, which is practically identical to the amount estimated in *E. coli* but twice as in JCVI-syn3A (Table EV1).

In cells, RNA can be subdivided into three major classes, i.e., rRNA, tRNA, and mRNA. In both bacteria and eukaryotes, rRNA constitutes the predominant form of cellular RNA, representing approximately 80% of the total RNA mass (Westermann *et al*, 2012; Bionumbers, 2015). Prokaryotes rRNA is composed of the 5S, 16S, and the 23S rRNA, which are typically organized as a co-transcribed operon and produced by the cleavage of a long precursor transcript. In *M. florum*, two copies of the rRNA locus are present in the genome. Our 5′-RACE results confirmed that *M. florum* rRNA genes are indeed transcribed as single polycistronic transcripts corresponding to TU_090 and TU_229 (Datasets EV3 and EV5). The remaining proportion of cellular RNA is composed of tRNA (~ 15%), mRNA (~ 5%), and other less abundant species such as sRNA and asRNA (< 1%) (Westermann *et al*, 2012). According to our macromolecular quantification results (see Table 1) and supposing that the proportions of RNA classes are conserved in *M. florum*, rRNA, tRNA, and mRNA have a total mass of 3.91, 0.73, and 0.24 fg, respectively (Dataset EV5). If we assume that the 5S, 16S, and 23S rRNAs are found at equimolar ratios, the calculated rRNA mass and estimated molecular weight suggest that roughly 4,900 rRNA molecules are present in a single *M. florum* cell (see Dataset EV5). Using the same assumption for tRNA species, approximately 18,000 tRNA molecules would also be present. Given the most probable *M. florum* cell volume (Table 1), this means that rRNAs and tRNAs would be found at a concentration of ~ $5.4 \times 10^4$ rRNAs/μm$^3$ and ~ $2.0 \times 10^5$ tRNAs/μm$^3$, respectively (Table EV1). tRNAs would thus be almost four times more abundant than rRNA molecules even though they occupy only ~ 15% of the total RNA mass.

We next used our RNA-seq data to estimate the intracellular abundance of each *M. florum* mRNA species (Dataset EV5). We observed excellent correlations between replicates (average Pearson correlation of 0.91) when considering the number of fragments per kilobase per million of mapped reads (FPKM) calculated for all *M. florum* CDS (Appendix Fig S5B). The FPKM values averaged over all replicates followed a typical Poisson distribution, with two-thirds of CDS (453/685) siting between 0 and 1,000 FPKM (Fig 5A and Appendix Fig S5C). A total of 660 CDS showed a detectable expression level (FPKM > 0), and 314 of these were expressed at a higher level than if all the reads were equally distributed across the

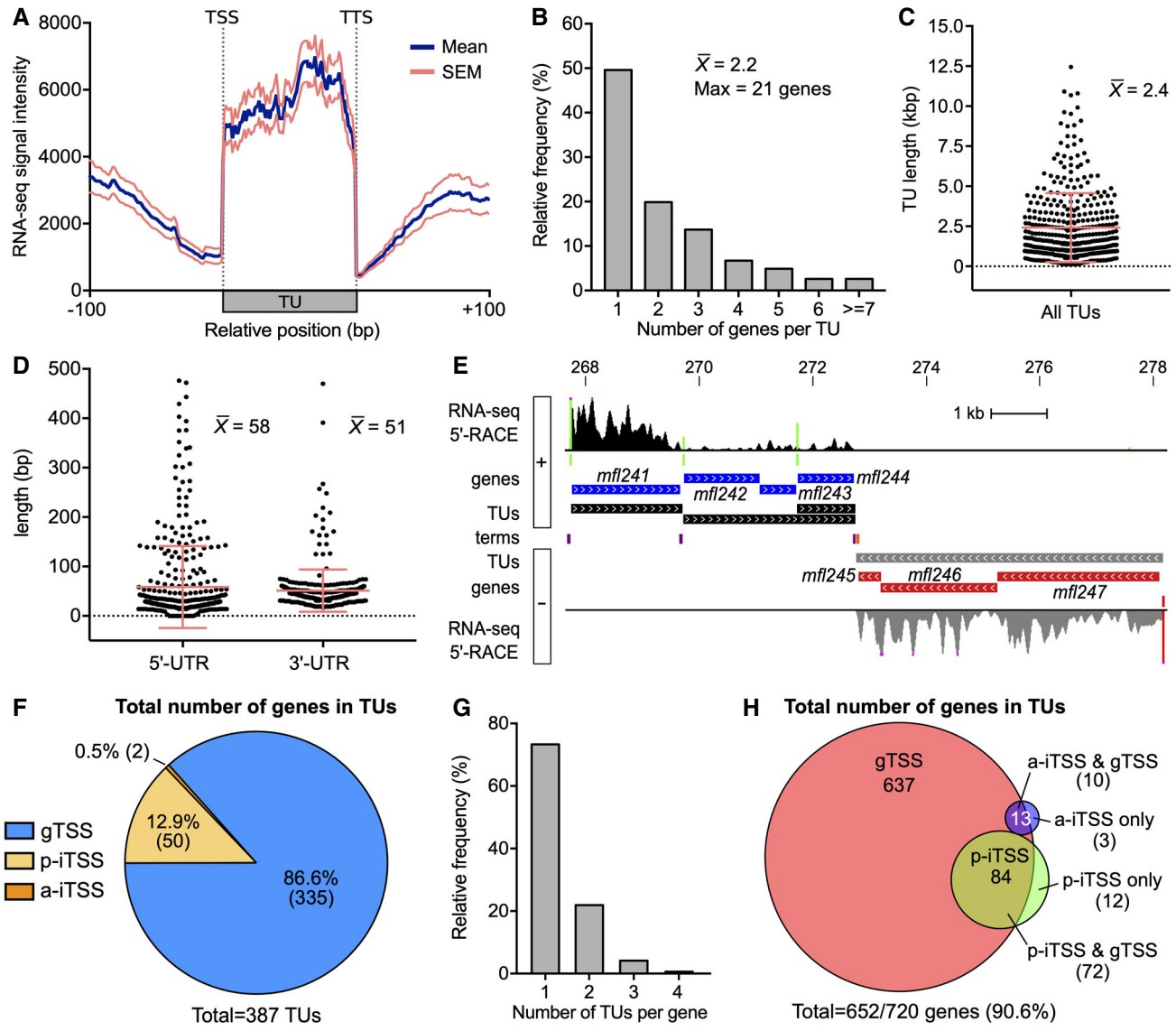

**Figure 4.  Analysis of reconstructed *M. florum* transcription units (TUs).**

A  Aggregate profile showing the mean RNA-seq read coverage observed for all reconstructed TUs and their surrounding DNA regions. The calculated SEM is also shown. The aggregate profile was centered on the TUs start and stop coordinates, corresponding to transcription start site (TSS) and termination site (TTS), respectively.

B  Relative frequency distribution of the number of genes per TU. The average and the maximal number of genes per TU are indicated.

C  Scatter plot showing the length of all reconstructed TUs. The mean and associated SD are shown.

D  Scatter plot showing the 5′ and 3′ untranslated regions (UTR) length of reconstructed TUs. The mean and associated SD are shown for each UTR type.

E  Genomic locus showing a representative example of reconstructed TUs. Genomic coordinates are indicated at the top of the panel (kb). From innermost to outermost tracks: terminators predicted on the positive (purple) and negative (orange) DNA strands; coordinates of TUs on the positive (black) and negative (gray) DNA strands; *M. florum* genes encoded on the positive (blue) and negative (red) DNA strands; position of motif-associated TSSs identified on the positive (green) and negative (red) DNA strands; RNA-seq and 5′-RACE signals observed on the positive and negative DNA strands, colored-coded identically to TUs and identified TSSs, respectively. Illustrated RNA-seq and 5′-RACE signals represent the number of read and read starts observed for a given position, respectively. RNA-seq signal was smoothed using a 5 pixels window (UCSC Genome Browser integrated function). RNA-seq and 5′-RACE peaks above 20,000 reads and 1,000 read starts are cut and marked by fuchsia dots, respectively.

F  Proportion of TUs per TSS type. a-gTSS are by definition excluded from the analysis since they are facing the nearest downstream gene.

G  Relative frequency distribution of the number of TUs per *M. florum* gene.

H  Venn diagram showing the total number of genes included in TUs generated from the different TSS types.

*M. florum* genome (FPKM > 630) (Fig 5A and Appendix Fig S5D). Many metabolic genes involved in glycolysis showed particularly high expression levels, notably peg.600 (*mfl596*; L-lactate

dehydrogenase), peg.583 (*mfl578*; glyceraldehyde-3-phosphate dehydrogenase), and peg.582 (*mfl577*; phosphoglycerate kinase) (Fig 5A and Dataset EV5). Interestingly, three of the ten most

expressed genes were annotated as hypothetical proteins, suggesting that important cellular functions are still unidentified in the current genome annotation. We also observed a striking difference in the transcription levels of CDS included in TUs compared with orphan CDS, which displayed significantly lower expression values (Fig 5B). However, we did not observe any clear correlation between the TSS signal intensity of a TU and the expression of its associated genes. According to the measured RNA mass (Table 1) and calculated

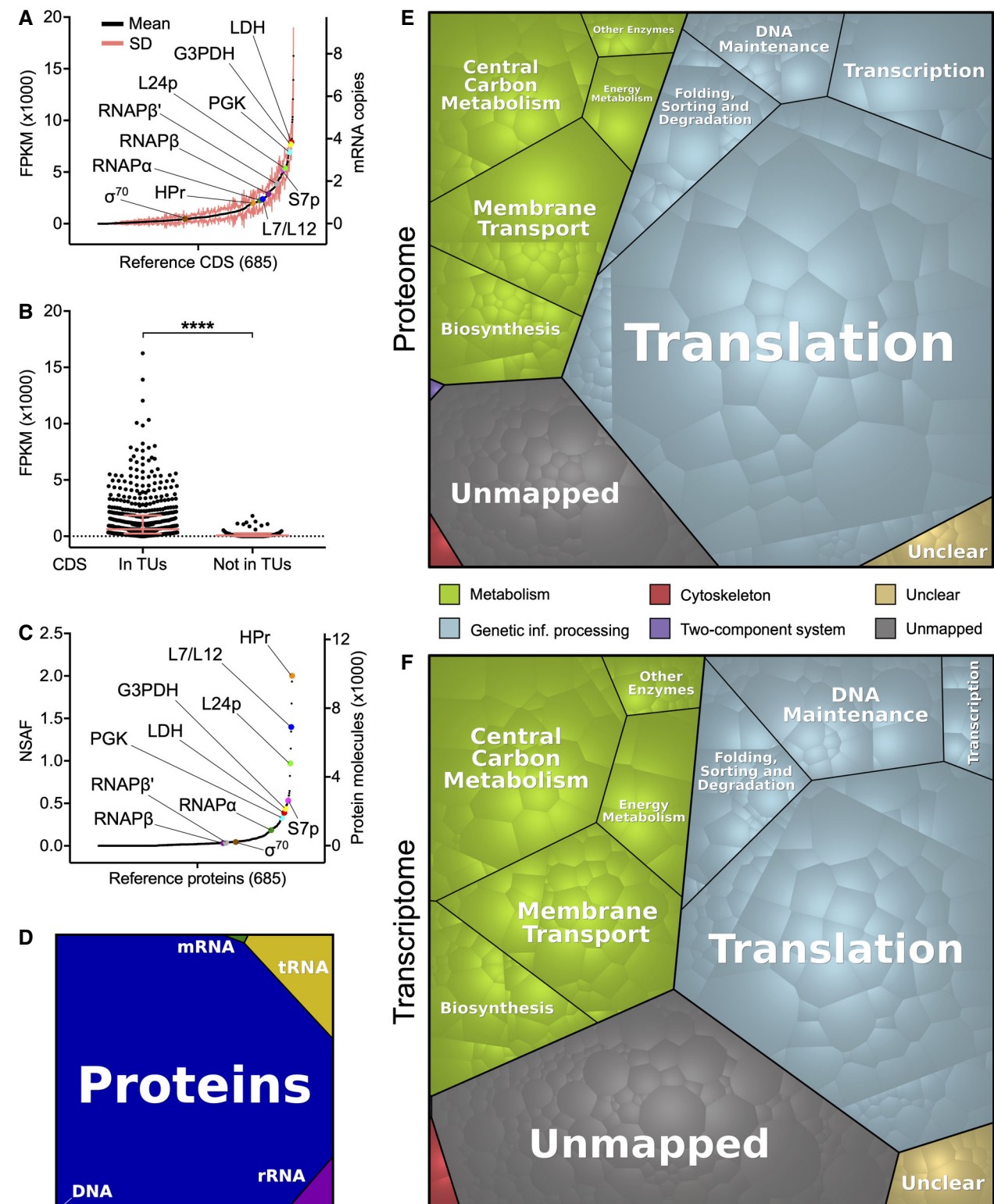

**Figure 5.**

**Figure 5. Expression levels of *M. florum* protein-coding genes and enrichment of functional categories.**

A Transcription levels of all *M. florum* coding sequences (CDS) quantified by RNA-seq. Transcription levels were calculated according to the number of fragments per kilobase per million of mapped reads (FPKM) observed over six replicates. The corresponding numbers of mRNA copies per cell, estimated from the measured *M. florum* RNA mass, are also indicated. CDS were sorted from least to most transcribed. The transcription level of selected genes of importance is presented. LDH, L-lactate dehydrogenase (peg.600/*mfl596*); G3PDH, glyceraldehyde-3-phosphate dehydrogenase (peg.583/*mfl578*); PGK, phosphoglycerate kinase (peg.582/*mfl577*); L24p and L7/L12, large subunit ribosomal proteins L24p (peg.133/*mfl134*) and L7/L12 (peg.605/*mfl601*); S7p, small subunit ribosomal protein S7p (peg.626/*mfl623*); RNAPβ, RNAPβ′, and RNAPα, RNA polymerase subunits β, β′, and α (peg.601/*mfl597*, peg.602/*mfl598*, and peg.149/mfl150); HPr, phosphotransferase system phosphocarrier protein HPr (peg.570/*mfl565*); σ$^{70}$, RNA polymerase sigma factor RpoD (peg.269/*mfl270*).

B Transcription level of CDS included in transcription units (TUs) compared with CDS not attributed to any TU (orphan CDS). The median and interquartile range are shown for both groups. The mean rank of each group was compared using a Mann–Whitney test (two-sided, ****P-value < 0.0001).

C Expression levels of all *M. florum* reference proteins quantified by two-dimensional liquid chromatography-tandem mass spectrometry (2D LC-MS/MS). Abundance was estimated according to the normalized spectral abundance factor (NSAF) calculated for each protein. A NSAF value of 0 was assigned to undetected proteins. The corresponding number of protein molecules per cell (derived from the biomass data) is indicated. Proteins were sorted from least to most abundant. The selected genes of importance presented in panel A are also highlighted.

D Overall DNA, tRNA, rRNA, mRNA, and protein proportions in terms of intracellular abundances in *M. florum*.

E Voronoi diagram illustrating the relative abundance of *M. florum* reference proteins grouped into different functional categories. Each polygon represents a specific protein weighted by its expression level quantified by 2D LC-MS/MS. Functions were attributed based on the KEGG Orthology (KO) database (Kanehisa et al, 2016a). The unmapped category regroups proteins for which no KO identifier could be assigned, while the unclear category contains proteins with KO numbers matching to unclear functions.

F As panel E but for mRNA abundances quantified by RNA-seq.

FPKM values, we estimated that a total of approximately 420 mRNA molecules are expected to be present at any moment within an exponential-phase *M. florum* cell growing in rich medium (Dataset EV5). If we normalize this value according to the most probable *M. florum* volume (Table 1), this represents ~ $4.7 \times 10^3$ mRNAs per μm$^3$ of cell volume, which is similar to numbers found in *M. pneumoniae* and *E. coli* (Table EV1). The expression value of most CDS (553/685) corresponded to less than one mRNA copy per cell, suggesting heterogenous expression levels between cells of the population and dynamic control of gene expression. Considering that *M. florum* has a doubling time of approximately 32 min (see Fig 1C) and that most bacterial mRNA have a very short half-life (less than 7 min for *Bacillus subtilis* (Hambraeus et al, 2003) or between 3 and 8 min for *E. coli* (Bernstein et al, 2002)), it is fair to assume that the entire *M. florum* mRNA pool of is almost completely renewed after one generation. In fact, more than 1,000 mRNA molecules are expected to be synthetized during a single-cell cycle. mRNA transcribed at less than one copy per cell could thus be expressed at substantial levels at some points during the cell cycle. For mRNAs that may not be expressed at each cycle, the corresponding proteins could still exert their functions over many generations since the half-life of bacterial proteins is typically ~ 20 h (Levy & Koch, 1955; Borek et al, 1958; Maier et al, 2011).

We previously showed that proteins occupy nearly half (46.6%) of the total *M. florum* dry mass (Fig 2F and Table 1). However, this macromolecular quantification did not provide information about the identity and specific abundance of the different proteins produced by the cell, which is highly relevant in the context of whole-cell modeling approaches such as GEMs. We therefore performed two-dimensional liquid chromatography–tandem mass spectrometry (2D LC-MS/MS) on an exponential-phase *M. florum* culture and analyzed the resulting spectra using three different search engines to maximize the identification of peptides matching the genome annotation (see Materials and Methods). More than 6,400 unique validated peptides were identified, altogether supported by more than 40,000 validated spectra at 1% false-discovery rate (FDR). Both the validated peptides and matching spectra showed very high average confidence rates (98.9%). More importantly, the detected peptides matched with 481 different *M. florum* ORFs, with each corresponding protein supported by an average of 84.3 peptides (13.2 validated peptides), for a mean protein coverage of ~ 33.0% (Dataset EV6). For 402 out of the 481 detected proteins (~ 84%), the region immediately upstream the corresponding ORF contained a ribosome binding site motif very similar to the Shine-Dalgarno consensus sequence (Fig EV5). The detected proteins also showed a very high average confidence rate (99.8%), and similarly to the estimated transcription levels, the normalized spectral abundance factor (NSAF) associated with each protein followed a Poisson distribution (Fig 5C and Dataset EV6). Indeed, a very low numbers of proteins, mainly ribosomal proteins, were detected at strikingly high levels, while most proteins showed medium to relatively low expression levels. Nonetheless, the correlation between transcription (FPKM) and protein expression (NSAF) levels was relatively modest (Spearman r = 0.61), a tendency also observed in other organisms (Greenbaum et al, 2003; Maier et al, 2009; Yang et al, 2014; Mazin et al, 2014; Kuchta et al, 2018). Using the molecular weight calculated for each protein and the total protein mass (Table 1), we converted the associated NSAF into absolute molecular quantities. According to our data, the average *M. florum* cell should contain approximately 250,000 protein molecules, with the most abundant protein present at almost 10,000 copies (peg.570/Mfl565, HPr PTS phosphocarrier protein) (Fig 5C and Dataset EV6). This represents more than ten times more molecules compared with the RNA fraction of the cell, for roughly twice the mass (Fig 5D). If we normalize the number of protein molecules per unit of cell volume, this represents roughly $2.8 \times 10^6$ proteins/μm$^3$, which is comparable to protein concentrations reported for JCVI-syn3A, *M. pneumoniae*, and *E. coli* (Table EV1).

## Overview of expressed cellular functions

Finally, we used our proteomic quantification data to visualize what cellular functions were predominantly expressed by *M. florum*. We therefore assigned KEGG Orthology (KO) identifiers (Kanehisa et al, 2016a) to *M. florum* reference ORFs and retrieved the associated functional categories. A KO number was successfully attributed to a

total of 435 *M. florum* proteins, of which 22 showed unclear function (Dataset EV7). Since the same protein can be assigned to multiple functional categories, we then curated the assigned categories based on the non-redundant Proteomap functional hierarchy (Liebermeister *et al*, 2014). This allowed the creation of a curated tree-like functional hierarchy for 413 different *M. florum* annotated proteins (Table 2 and Dataset EV7). The predicted functions of these proteins could be regrouped in just 27 different functional categories, illustrating the striking simplicity of this organism. We then used weighted Voronoi diagrams to visualize the relative importance of the assigned functional categories (Liebermeister *et al*, 2014). Unsurprisingly, the largest portion of the *M. florum* proteome was occupied by proteins implicated in translation processes, representing almost half (49.0%) of the total protein molecules of the cell and 33.5% of the protein mass (Fig 5E, Datasets EV6 and EV7). Central carbon metabolism and membrane transport categories also displayed particularly important proteome fractions, accounting for 7.5 and 7.4% of the *M. florum* protein diversity, respectively (Fig 5E). On the other hand, only very limited proteome allocation (< 1%) was devoted to cytoskeleton and two-component system functional categories. More importantly, proteins assigned to functional categories (excluding the unclear function category) comprised 86.0% of the total estimated protein molecules per cell, representing 82.1% of the *M. florum* protein mass (Fig 5E, Datasets EV6 and EV7). Functional categories weighted with the estimated mRNA abundances also showed the same overall picture, with however a slightly larger portion occupied by metabolism and unmapped categories (Fig 5F). Additional experiments would be required to determine the role of proteins with unknown or hypothetical function, and therefore assign the remaining protein fraction to the appropriate functional categories. Interestingly, our protein quantification data and functional category assignments can be used to estimate the abundance of conserved protein complexes, the bacterial ribosome for example. According to our analysis, we estimated that each *M. florum* cell should contain between 1,600 and 2,100 ribosomes. This corresponds to approximately 18,000 to 24,000 ribosomes per $\mu m^3$ of cell volume, concentrations in range with values reported for *M. mycoides* and *E. coli* (Table EV1). We also estimated that ~ 270 core RNA polymerase (RNAP) should be present in the average *M. florum* cell (~ 3,000 RNAP/$\mu m^3$), which nearly matches the number of $\sigma^{70}$ factor per cell (~ 230).

## Discussion

Due to its interesting characteristics, *M. florum* is an attractive model organism for synthetic genomics and systems biology. This near-minimal bacterium possesses a genome smaller than those of most current model organisms (e.g. *E. coli*, *M. pneumoniae*, *M. mycoides*), grows rapidly in standard laboratory conditions, and is classified as a BSL-1 organism. The flip side of being non-pathogenic is that until recently, only little attention had been given to *M. florum*, although it was isolated almost 40 years ago (McCoy *et al*, 1980; Whitcomb *et al*, 1982; McCoy *et al*, 1984). Consequently, practically no quantitative data about the physiology of *M. florum* was available in the literature, and many important aspects of its cellular mechanisms and metabolism remained uncharacterized. Here, we measured several physical, physiological,

and molecular characteristics of *M. florum* and integrated the generated data to estimate parameters difficult to evaluate using conventional laboratory equipment. A summary of the characterization reported in this study is presented in Fig 6. More specifically, we precisely evaluated the *M. florum* growth kinetics in rich medium (Fig 1) and measured the cell diameter, buoyant density, and dry mass to infer the most probable cell mass, volume, and surface area (Fig 2 and Table 1). We also quantified the macromolecular mass fractions of the cell (Figs 2F and EV1) and proceeded to the first experimental cartography of *M. florum* TUs based on 5′-RACE TSSs identification results and Rho-independent terminator predictions (Figs 3 and 4, and EV2–EV4, Appendix Figs S4–S10, and Datasets EV1–EV4). Finally, we quantified the transcription and protein expression levels of all *M. florum* reference CDS, used the macromolecular quantification results to estimate absolute mRNA and protein abundances, and exploited these estimations to evaluate the relative importance of protein functional categories (Fig 5, Table 2, and Datasets EV5–EV7).

While *M. florum* has never been associated with any disease, this does not completely rule out the possibility that this bacterium could be pathogenic in yet unidentified circumstances or for specific organisms. However, since the growth of *M. florum* L1 is impaired at 36°C and completely abolished at 38°C, the probability that it infects warm-blooded animals is very low. In addition, no known virulence factor is predicted from its genome sequence. The exact nature of the primary niche of this bacterium remains unclear, but the previous isolation of various *M. florum* strains from insects suggests that it could be a commensal of the digestive tract of these organisms (Tully *et al*, 1987; Baby *et al*, 2018). This is further supported by the fact that many members of the *Entomoplasmatales* group, including several species of the *Mesoplasma*, *Spiroplasma* and *Entomoplasma* genera, have been isolated from or are associated with arthropods (Tully *et al*, 1993; Tully *et al*, 1994; Funaro *et al*, 2011; Brown & Bradbury, 2014; Sapountzis *et al*, 2018). This would also explain the presence of *M. florum* on plants (McCoy *et al*, 1980, 1984; Whitcomb *et al*, 1982; Baby *et al*, 2018). The digestive tract of insects would provide a unique environment in which *M. florum* would have continuous access to complex nutrients such as lipids and peptides to palliate for its metabolic deficiencies, as well as to various sugar sources depending on the diet of its host. Additional data are however required to confirm this hypothesis.

In our growth experiments, *M. florum* exhibited the four typical bacterial growth phases (Fig 1B–D). The measured $OD_{560 \, nm}$ signal, which was shown to correlate with the growth medium pH (Matteau *et al*, 2015), showed a progressive acidification during exponential phase. Since the main route for energy production in *M. florum* is predicted to be the glycolysis pathway and that no tricarboxylic acid (TCA) cycle is present, this gradual acidification is most probably caused by the accumulation of fermentation products (lactate and acetate) in the medium (Pollack *et al*, 1997; Halbedel *et al*, 2007; Caspi *et al*, 2014, 2016). The decrease in $OD_{560 \, nm}$ eventually reached a plateau, corresponding to a medium pH of ~ 6.0, which also coincided with the beginning of the death phase. At that point, the high concentration of protons in the medium is most likely toxic, but the underlying mechanisms resulting in *M. florum* death remain unknown. Compared with most Mollicutes, *M. florum* showed a remarkably fast doubling time (~ 32 min) (Fig 1C). For example, *M.*

**Table 2.** Curated functional hierarchy tree of *Mesoplasma florum* annotated ORFs.

| Functional category | Subcategory | Sub subcategory | Number of ORFs | % of total ORFs |
|---|---|---|---|---|
| Cellular processes | Cytoskeleton | Cytoskeleton proteins | 2 | 0.3 |
| Environmental information processing | Signal transduction | Two-component system | 1 | 0.1 |
| Genetic information processing | DNA maintenance | DNA repair | 23 | 3.4 |
| | | DNA replication and partition | 30 | 4.4 |
| | | Subtotal | 53 | 7.7 |
| | Folding, sorting and degradation | Chaperones and folding catalysts | 7 | 1.0 |
| | | Nucleases | 11 | 1.6 |
| | | Peptidases | 9 | 1.3 |
| | | Protein export | 7 | 1.0 |
| | | Sulfur relay system | 2 | 0.3 |
| | | Subtotal | 36 | 5.3 |
| | Transcription | RNA polymerase | 5 | 0.7 |
| | | Transcription factors | 6 | 0.9 |
| | | Subtotal | 11 | 1.6 |
| | Translation | Ribosome | 51 | 7.4 |
| | | Ribosome biogenesis | 29 | 4.2 |
| | | Translation factors | 11 | 1.6 |
| | | tRNA loading and maturation | 30 | 4.4 |
| | | Subtotal | 121 | 17.7 |
| | Total | | 221 | 32.3 |
| Metabolism | Biosynthesis | Amino acid metabolism | 5 | 0.7 |
| | | Cofactor biosynthesis | 16 | 2.3 |
| | | Lipid and steroid metabolism | 8 | 1.2 |
| | | Purine and pyrimidine metabolism | 23 | 3.4 |
| | | Subtotal | 52 | 7.6 |
| | Central carbon metabolism | Glycolysis and carbohydrate metabolism | 35 | 5.1 |
| | | Other central metabolism enzymes | 6 | 0.9 |
| | | Pentose phosphate metabolism | 8 | 1.2 |
| | | Subtotal | 49 | 7.2 |
| | Energy metabolism | Oxidative phosphorylation | 9 | 1.3 |
| | Membrane transport | PTS system | 13 | 1.9 |
| | | Secretion system | 2 | 0.3 |
| | | Transport | 42 | 6.1 |
| | | Subtotal | 57 | 8.3 |
| | Other enzymes | Other enzymes | 22 | 3.2 |
| | Total | | 189 | 27.6 |
| Not mapped | – | – | 250 | 36.5 |
| Unclear | – | – | 22 | 3.2 |
| Grand total | | | 685 | 100.0 |

*mycoides* subspecies *capri* exhibits a doubling time of ~ 60 min in rich medium (Gibson *et al*, 2010; Hutchison *et al*, 2016), while it is estimated to be around 90 min for *M. capricolum* subspecies *capricolum* (Seto & Miyata, 1998) and 8–20 h for *M. pneumoniae* (Yus *et al*, 2009; Wodke *et al*, 2013). Intriguingly, *M. genitalium*, which possesses the smallest genome among all Mollicutes (~ 580 kb), has an extremely slow growth rate corresponding to a doubling time of ~ 16 h (Jensen *et al*, 1996; Hutchison *et al*, 2016). Clearly, the

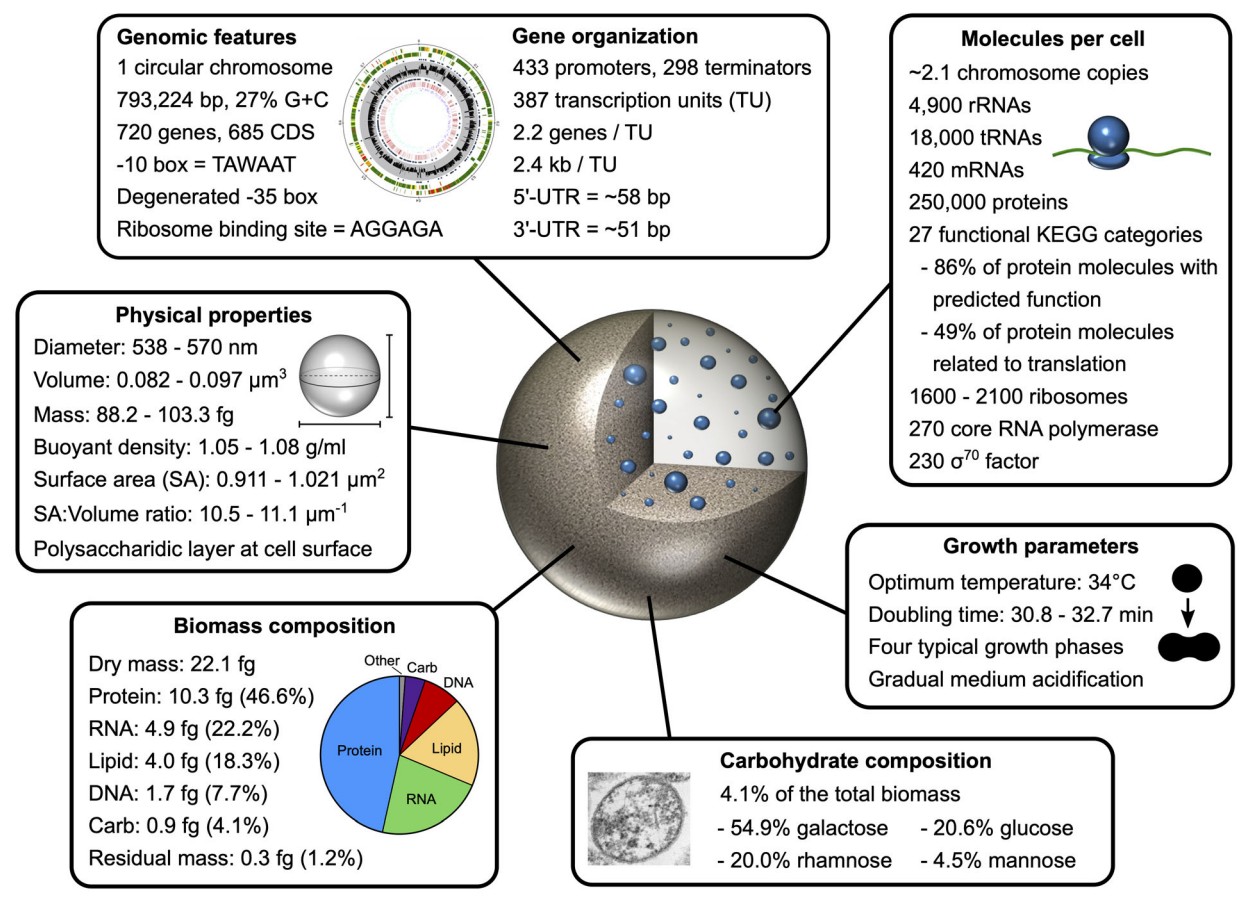

**Figure 6. Overview of the *M. florum* characterization reported in this study.**

doubling time of Mollicutes is not correlated with their genome size, and the factors contributing to a fast-growing phenotype are still elusive. This trait is most likely related to the selective pressures and evolutionary strategies adopted by specific species in their natural habitat. The utilization of a GEM that integrates the metabolic fluxes, nutrients availability, growth rate, and ATP production rate of *M. florum*, and more importantly its comparison with other Mollicutes GEMs, might yield more specific hypotheses on the underlying genetic factors contributing to the fast-growing phenotype of *M. florum*.

Using TEM and STED microscopy, we measured an average *M. florum* cell diameter of 434 and 741 nm, respectively (Fig 2A–D). This range of cell diameter was refined to 538–570 nm using a mathematical approach that integrates other physical parameters such as the buoyant cell density and the cell dry mass (Fig 2E–G). Overall, *M. florum* cells are slightly bigger than the reported size of *M. mycoides* subspecies *capri*, JCVI-syn1.0 and JCVI-syn3A (~ 400 nm) (Gibson *et al*, 2010; Breuer *et al*, 2019), but within typical ranges observed for most Mollicutes (~ 200–600 nm). More importantly, the determination of the most probable cell diameter allowed us to estimate the cell mass, volume, surface area, and *SA:V* ratio of *M. florum* (Table 1). According to our analysis, *M. florum* is expected to have a volume between 0.082 and 0.097 $\mu m^3$ during the exponential phase, which is nearly 50 times smaller than

*E. coli* growing in similar conditions (~ 4 $\mu m^3$) (Volkmer & Heinemann, 2011; Dai & Zhu, 2018). This important difference in cell volume is also apparent in the respective *SA:V* ratio of the two bacteria, with values approaching 10 $\mu m^{-1}$ for *M. florum* compared with ~ 4 $\mu m^{-1}$ for *E. coli*. Recent publications showed that bacteria exhibit robust *SA:V* ratio homeostasis in response to different types of perturbations, including nutritional shifts and genetic alterations (Harris & Theriot, 2016, 2018; Ojkic *et al*, 2019). Since Mollicutes have lost the ability to synthetize many important metabolites, their high *SA:V* ratios could represent a physical adaptation to increase their capacity of importing complex nutrients from the environment. Interestingly, this difference in *SA:V* ratios between *M. florum* and *E. coli* is also apparent when comparing the macromolecular mass fractions associated with each bacterium (Fig 2F). In *M. florum*, we showed that approximately 18% of the dry mass comes from lipids and 47% from proteins, whereas these fractions typically represent ~ 9 and ~ 55% of the *E. coli* dry mass (Dennis & Bremer, 1974; Feist *et al*, 2007; Bionumbers, 2015).

According to our TEM pictures and biomass quantification results (Fig 2A and F), *M. florum* produces a surface polysaccharide layer primarily composed of galactose (54.9%) and glucose (20.6%). This suggests the presence of a biosynthesis pathway similar to what is found in *M. mycoides* and *M. capricolum* (Razin *et al*, 1963; Bertin *et al*, 2013; Gaurivaud *et al*, 2014; Daubenspeck *et al*,

2014; Bertin *et al*, 2015). However, the genetic determinants responsible for the synthesis of this polysaccharidic layer, its biological function, and the precise organization of its sugar monomers remains to be identified in *M. florum*. Additionally, it is still unclear whether this thin layer constitutes capsular polysaccharides (CPS) covalently bound to the cell surface or exopolysaccharides (EPS) secreted in the culture medium that passively coat *M. florum* cells. In fact, both forms could exist and be subjected to regulation depending on environmental conditions or specific signals. Since *M. florum* cells were washed several times prior biomass quantification and TEM examination, the sole presence of EPS would be unlikely. In the environment, this layer could potentially serve as a protection against desiccation outside of its host. This would provide increased survivability on plant surfaces and contribute to its dissemination across insect populations. An important proportion (20.0%) of the *M. florum* carbohydrate mass also consisted of rhamnose, a monosaccharide commonly found in mycoplasmas and involved in the attachment of proteins on the cell membrane (Jordan *et al*, 2013; Daubenspeck *et al*, 2016). This anchoring process is thought to provide cytoplasmic proteins with additional functions, giving them the ability to moonlight on the cell surface. The considerable amount of rhamnose present in *M. florum* biomass indicates that this mechanism could also play a role in this species.

In this study, we combined 5′-RACE and RNA-seq methodologies to draw a first portrait of the *M. florum* transcriptome. The analysis of 5′-RACE reads revealed 432 TSSs associated with a promoter motif sharing important similarities with previously characterized Mollicutes promoters, including a highly conserved Pribnow box (TAWAAT), a partially conserved EXT element, and a highly degenerated −35 box (Fig 3B, E and F, and Dataset EV1) (Weiner III, 2000; Güell *et al*, 2009; Weber *et al*, 2012; Yus *et al*, 2012; Mazin *et al*, 2014; Lloréns-Rico *et al*, 2015; Fisunov *et al*, 2016). Since no other motif could be found and that only one σ factor is predicted in *M. florum* (σ$^{70}$), this promoter motif is most likely responsible for the transcription of nearly all *M. florum* genes. Overall, these observations strengthen once again the idea that the −35 box and the EXT element could be less important for promoter recognition in Mollicutes compared with other bacteria such as *B. subtilis* or *E. coli*. In *M. gallisepticum*, for instance, only 122 mapped TSSs (out of 1,061) were shown to be associated with a −35 box motif (Mazin *et al*, 2014), while in *M. pneumoniae*, attempts to determine a clear −35 box motif were apparently unsuccessful (Weiner III, 2000; Güell *et al*, 2009; Lloréns-Rico *et al*, 2015). The EXT element was also shown to be absent from the core promoter of *M. gallisepticum* and *M. pneumoniae*, whereas it appears to be fairly conserved in *M. hyopneumoniae* and in *A. laidlawii* (Weiner III, 2000; Güell *et al*, 2009; Weber *et al*, 2012; Yus *et al*, 2012; Mazin *et al*, 2014; Lloréns-Rico *et al*, 2015; Fisunov *et al*, 2016). In some cases, the EXT element could compensate the absence of the −35 box as previously demonstrated with *B. subtilis* and *Streptococcus pneumoniae* (Sabelnikov *et al*, 1995; Voskuil & Chambliss, 1998). Still, many Mollicutes promoters seem to rely only on the −10 box to properly interact with the RNA polymerase and initiate transcription at the +1 site. Other regions such as A-T rich region located between the −35 position and the EXT element might play a role in promoter recognition and in the formation of the open promoter complex. High-throughput approaches using randomized promoter libraries could be an efficient strategy to analyze the importance of promoter elements

and explore the diversity of sequence enabling transcription initiation in *M. florum* (Mutalik *et al*, 2013; Guiziou *et al*, 2016).

In A-T rich genomes, the number of spurious Pribnow boxes arising at unexpected genomic positions such as within coding regions is expected to be particularly high (Lloréns-Rico *et al*, 2016; Wade & Grainger, 2018). These cryptic elements contribute to a genome-wide and low level transcriptional noise, a phenomenon referred as pervasive transcription (Wade & Grainger, 2014, 2018). Interestingly, our 5′-RACE data revealed 181 putative TSSs, mostly located within coding regions of the genome, which could not be associated with the identified *M. florum* promoter motif (Figs 3A and C, and EV2A). Additional efforts to search for promoter sequence similarities among these TSSs were unsuccessful. These 5′-RACE peaks are probably the result of low affinity-binding events of the σ$^{70}$ subunit to sequences faintly resembling to promoter elements, resulting in the initiation of transcription at spurious sites. However, since the intensity of these TSSs is globally very low (Fig EV2B), the energetic cost related to the synthesis of the associated transcripts as well as their potential impact on the normal transcription of overlapping genes is most likely negligible (Lloréns-Rico *et al*, 2016). Even though pervasive transcription seems to be widespread across bacterial species (Dornenburg *et al*, 2010; Chao *et al*, 2012; Nicolas *et al*, 2012; Lybecker *et al*, 2014; Mazin *et al*, 2014; Haycocks & Grainger, 2016; Lloréns-Rico *et al*, 2016), its putative biological function remains controversial. Spurious promoters might in fact serve as a reservoir on which natural selection can operate to produce functional transcripts such as sRNAs and asRNAs, thus participating to the overall transcriptome plasticity of cells (Jose *et al*, 2019). We indeed observed that a small proportion (22%) of identified motif-associated TSSs were located within coding regions of the *M. florum* chromosome (iTSSs) (Fig 3D and Dataset EV1), and many of them could not be attributed to any downstream gene (non-coding TUs and orphan TSSs), suggesting the presence of sRNAs or asRNAs (Dataset EV4). Motif-associated iTSSs were however characterized by weaker associated 5′-RACE and RNA-seq signal intensities compared with intergenic TSSs (gTSSs) (Fig EV2B, Appendix Figs S6 and S10). Many of these putative transcripts might be only expressed at substantial levels under specific conditions or stresses, as observed in other bacteria (Dornenburg *et al*, 2010; Chao *et al*, 2012; Nicolas *et al*, 2012; Lybecker *et al*, 2014; Mazin *et al*, 2014; Haycocks & Grainger, 2016; Lloréns-Rico *et al*, 2016). In some instances, these transcripts could even encode for alternative open reading frames (AltORFs) (Vanderperre *et al*, 2013; Mouilleron *et al*, 2016) or small ORFs (≤ 100 amino acids) (Lluch-Senar *et al*, 2015; Ravikumar *et al*, 2018; Miravet-Verde *et al*, 2019). The analysis of mass spectrometry data using a six-frame translated database could provide significant evidences in that context.

Using the identified motif-associated TSSs and the predicted Rho-independent terminators, we reconstructed 387 TUs in *M. florum*, encompassing more than 90% of all annotated genes (Fig 4, Appendix Fig S8, and Dataset EV3). Since many motif-associated iTSSs were properly disposed to drive the expression of downstream genes (Figs 3D and EV4) and displayed very similar characteristics compared with gTSSs (Fig 3E–G), these TSSs were also included in the reconstruction of *M. florum* TUs (Fig 4F). Although about half of TUs were shown to contain only a single gene, many TUs were polycistronic, and about 25% of *M. florum* genes were included in more than one TU (Fig 4B and G). This resulted in a surprisingly

complex transcriptome architecture comparable to previous characterizations conducted in *M. pneumoniae* and *M. gallisepticum* (Güell *et al*, 2009; Mazin *et al*, 2014), with many overlapping TUs and an important fraction of genes apparently transcribed from multiple promoters. Curiously, the majority of genes located downstream of iTSSs were apparently also transcribed from a gTSS (Fig 4 H). In fact, of the 15 genes strictly transcribed from iTSSs, nine happened to be expressed from iTSSs located exactly on translation start codons (leaderless mRNA), leaving only six genes controlled by true internal promoters. The actual role of intragenic promoters in *M. florum* is puzzling. In some cases, they could simply be the results of acquired mutations that were not counter-selected because of the absence of any deleterious effect on the transcription of neighboring genes. In other situations, they could be important for the optimal expression of certain genes via the production of supplementary mRNA isoforms. Some of these promoters could actually constitute regulatory platforms for the biding of transcriptional factors modulating transcription upon specific signals. While our results demonstrate an impressive transcriptome complexity, our TU reconstructions were also based on the assumption that all predicted terminators were 100% efficient, which almost certainly underestimates the full transcriptome diversity in *M. florum*. Recent studies showed that transcription terminators are often not entirely efficient, allowing transcriptional readthrough and thus contributing to the transcription of downstream elements (Nicolas *et al*, 2012; Wade & Grainger, 2014; Lalanne *et al*, 2018). Nevertheless, our RNA-seq data correlated very well with the reconstructed TU boundaries as well as terminator predictions (Fig 4A, Appendix Figs S7 and S9), suggesting that transcriptional readthrough is not predominant in *M. florum*. Termination readthrough could still be responsible for the very low expression of genes not associated with any of the identified promoters, which represent roughly 10% of all *M. florum* genes (Fig 5B). Of course, as this represents the very first characterization of the *M. florum* transcriptome, it will be possible to integrate additional datasets to improve its overall precision and breadth. For example, methods that inform about the 3′ end coordinates of transcripts such as the Rend-seq (Lalanne *et al*, 2018) could be used to validate and improve the current terminator predictions, in addition to potentially highlight occurrences of leaky terminators.

Achieving a complete and quantitative description of all constituents of a cell represents one of the most important goals of systems biology. To understand global properties of complex biological systems such as cells, one must clearly identify and quantify their components. In this study, we estimated that the average *M. florum* cell contains approximately 250,000 proteins, 4,900 rRNAs, 18,000 tRNAs, 420 mRNAs, and 2.1 copies of the chromosome (Fig 5D, Table EV1, and Datasets EV5 and EV6). Considering the functional categories assigned by the KO database, we further estimated that about 1,600 to 2,100 ribosomes, 270 core RNAP, and 230 $\sigma^{70}$ factor are expected to be present in the average *M. florum* cell (Table EV1 and Datasets EV6 and EV7). Overall, the abundance of RNA and protein molecules per cell is comparable to estimates in other Mollicutes but roughly ten times lower compared with *E. coli*, which is not surprising considering the large difference in respective cell volumes (Table EV1). Still, among the two other Mollicutes species selected for comparison, *M. florum* shows the highest number of proteins and ribosomes per cell but also has the highest cell volume with almost three times more cytoplasmic space than *M. mycoides*

subspecies *capri* or JCVI-syn3A (Table EV1). Yet, *M. florum* and *E. coli* show very similar RNA and protein concentrations when normalized for cell volume. The total number of proteins and ribosomes per unit of volume is also very consistent between all the species compared, with the exception of *M. pneumoniae* that has the lowest concentration of proteins and nearly ten times less ribosomes per µm³ (Table EV1). This disparity between *M. pneumoniae* and *M. florum* is also apparent when comparing the relative importance of protein functional categories in each species, with *M. pneumoniae* displaying significantly reduced investments in translation processes at the benefit of other processes such as cell motility and cytoskeleton (Fig 5E) (Kühner *et al*, 2009; Liebermeister *et al*, 2014). Consistently, the overall RNA levels of *M. pneumoniae* are also remarkably lower compared with *M. florum* and *E. coli*. This is not surprising since *M. pneumoniae* has only one rRNA operon per genome vs. two and seven for *M. florum* and *E. coli*, respectively. These observations are in agreement with the important difference between the growth rate of *M. florum* (~ 32 min) and *M. pneumoniae* (~ 8–20 h), supporting the idea that *M. pneumoniae* is not optimized for biomass production but rather depends on more complex strategies for fitness and competition in its natural environment (Yus *et al*, 2009). Furthermore, GEM reconstruction for *M. pneumoniae* revealed that most of the energy produced by this pathogenic bacterium is used for maintenance tasks instead of growth, strongly contrasting with *M. mycoides* subspecies *capri* (JCVI-syn3A) for which the complete opposite was observed (Wodke *et al*, 2013; Breuer *et al*, 2019).

Since *M. florum* and JCVI-syn3A share similar numbers of ribosomes per unit of volume but have different doubling times (~ 32 min vs. ~ 60 min), it would be interesting to compare how they allocate their resources between growth and maintenance tasks. Other parameters such as the overall efficiency of the glycolysis pathway or the efficiency of the gene expression machinery could also play an important role in the difference observed between their respective growth rate. By reconstructing whole-cell models for *M. florum*, it will be possible to integrate the data generated in this study to investigate these questions and gain additional knowledge about the global cell functioning of this near-minimal bacterium. Moreover, since we reconstructed M. *florum* TUs, we now have the data required to use whole-cell modeling algorithms such as MinGenome (Wang & Maranas, 2018) to improve the initial genome reduction scenarios based on gene essentiality and conservation (Baby *et al*, 2018). MinGenome identifies all dispensable contiguous sequences in size descending order and preserves promoter regions needed for proper transcription of the retained genes (Wang & Maranas, 2018). The minimal genome designs inferred by this method could then be systematically analyzed using modeling approaches and compared with the synthetic minimal organism JCVI-syn3A to highlight differences in their genome composition and retained protein functions. While some differences can probably be attributed to culture medium compositions, many cases could constitute examples of non-orthologous gene displacement or divergent evolutionary strategies to compete in their natural habitat, thereby shedding light on some of the principles behind minimal genome plasticity. Interesting genome architectures emerging from these analyses could next be subjected to total DNA synthesis and assembly in yeast followed by transplantation into a recipient bacterium. If successful, the transplanted synthetic

genomes could be analyzed using the methods described in this study to potentially acquire new knowledge about genome design principles, which are currently lacking and restraining the rational design of synthetic organisms.

# Materials and Methods

### Bacterial strains and growth conditions

All experiments were performed using *M. florum* strain L1 (ATCC 33453) grown with shaking in ATCC 1161 medium (1.75% ($w/v$) heart infusion broth, 4% ($w/v$) sucrose, 20% ($v/v$) horse serum, 1.35% ($w/v$) yeast extract, 0.004% ($w/v$) phenol red, 200 U/ml penicillin G (Matteau *et al*, 2017) at a temperature of 34°C (unless stated otherwise).

### Doubling time measurement using colorimetric assays

Colorimetric assays used to measure *M. florum* doubling time were based on growth assays previously developed for spiroplasmas (Konai *et al*, 1996). Briefly, ATCC 1161 medium was inoculated with an exponential-phase *M. florum* preculture to obtain an initial concentration of ~ $1 \times 10^5$ CFU/ml. The inoculated medium was then diluted using twofold serial dilutions to obtain a total of four dilutions (1:1, 1:2, 1:4, and 1:8). Each dilution was transferred in triplicate into a 96-well microplate, and the plate was incubated with shaking at the desired temperature (30, 32, 34, 36 or 38°C) in a Multiskan GO microplate reader (Thermo Scientific). Bacterial growth was monitored by measuring the $OD_{560 \, nm}$ every 10 min for ~ 16 h. The metabolic activity of *M. florum* was previously shown to result in the acidification of the ATCC 1161 growth medium, causing changes in the absorbance of phenol red at 560 nm that correlate with the number CFUs (Matteau *et al*, 2015). To calculate doubling times, linear regressions ($R^2 > 0.999$) were traced on the linear portion of the $OD_{560 \, nm}$ curves, and the amount of time separating each dilution curve was calculated according to the linear regression equations.

### Growth kinetics assays

Growth kinetics assays were performed in triplicate by monitoring the cell concentration of three independent *M. florum* cultures using CFU and FCM counts. Briefly, ATCC 1161 medium was inoculated with an exponential-phase *M. florum* preculture to obtain an initial concentration of ~ $1 \times 10^5$ CFU/ml. Inoculated medium was incubated at 34°C with shaking for ~ 24 h in an orbital shaker incubator. Aliquots were harvested every ~ 2 h and the $OD_{560 \, nm}$ was immediately measured in duplicate using a Multiskan GO microplate reader (Thermo Scientific). CFUs were evaluated in duplicate by spotting serial dilutions of the aliquots (in PBS1×) on ATCC 1161 solid medium and counting colonies after an incubation of 24–48 h at 34°C. 37% ($w/v$) formaldehyde was then added and mixed to each dilution to obtain a final concentration of 1% ($w/v$), and the plate was incubated at room temperature (RT) for ~ 25 min. SYBR Green I (Invitrogen) dye was added to a final concentration of 1×, mixed, and samples were incubated again at RT for ~ 25 min. Cell concentration was finally measured in duplicate using a BD Accuri C6 Plus

flow cytometer (BD Biosciences) equipped with a 488 nm laser. FSC-H and FL1-H (FITC) channel thresholds were set at 100 and 750, respectively. Fluidics were set to high speed, and a maximum of 40 μl or $1 \times 10^6$ events were collected for each sample. We validated that cell concentrations were well correlated with culture dilutions diluted in PBS1× (Appendix Fig S2), and appropriate controls were performed (PBS1× without cells, unstained cells, etc.).

### Cell viability assay

Cell viability of *M. florum* was assessed by SYTO 9 and PI double staining (Boulos *et al*, 1999). *M. florum* cells were centrifuged at 10°C for 2 min at 21,100 × $g$, and washed once with cold PBS1×. Cells were centrifuged again and then resuspended in PBS1× containing 5 μM SYTO 9 (Molecular Probes) and 10 μg/ml PI (Biotium). Cells were stained at RT for ~ 20 min. A fixed-cells control was also performed by incubating a *M. florum* washed cell aliquot with 1% ($w/v$) formaldehyde at RT for ~ 25 min. Fixed cells were centrifuged at 10°C for 2 min at 21,100 × $g$, resuspended in PBS1× containing 0.1% ($v/v$) Triton X-100, and incubated at RT for 2 min. Cells were centrifuged again and finally resuspended in PBS1× containing 5 μM SYTO 9 (Thermo Fisher Scientific) and 10 μg/ml PI (Biotium). Samples were immobilized on agarose pad slides and examined by widefield fluorescence microscopy using an Axio Observer Z1 inverted microscope (Zeiss) equipped with an AxioCam 506 mono (Zeiss) camera and a 100×/NA1.4 Plan-Apochromat oil immersion objective. SYTO 9 and PI were excited and acquisitioned using GFP and Cy3 excitation/emission filters, respectively. Images were captured with Zeiss Zen 2.0 imaging software and analyzed using Fiji (Schindelin *et al*, 2012).

### Stimulated emission depletion microscopy

Stimulated emission depletion (STED) microscopy was performed using double-stained (membrane and DNA) *M. florum* cells. Briefly, an exponential-phase *M. florum* culture was centrifuged at 10°C for 2 min at 21,100 × $g$ and washed twice with cold electroporation buffer [272 mM sucrose, 1 mM HEPES (pH 7.4)]. Washed cells were then immobilized on a poly-L-lysine-coated glass slide (Poly-Prep Slide, Sigma-Aldrich) and incubated at RT for 5 min. Cells were washed on slide twice with PBS1×, and then stained, fixed, permeabilized, and stained again for 5 min each at RT using the following solutions (all reagents diluted in PBS1×, with two PBS1× washes between each step): (i) 0.5 μM mCLING-ATTO 647N-labeled dye (Synaptic Systems); (ii) 4% ($w/v$) formaldehyde and 0.2% ($w/v$) glutaraldehyde; (iii) 0.1% (v/v) Triton X-100; and (iv) 1/100 dilution (100×) of PicoGreen concentrate reagent (Molecular Probes). Cells were washed twice again with PBS1× and then finally mounted for STED microscopy using ProLong Diamond Mountant (Molecular Probes). Two-color STED microscopy was performed using a DMi8 STED microscope (Leica TCS SP8) equipped with a 100×/NA1.4 HC Plan-Apochromat CS2 oil immersion objective and operated with the LAS X imaging software (version 3.1.1.15751, Leica). mCLING-ATTO 647N and PicoGreen were excited using a pulsed white light laser set at 646 and 488 nm, respectively, and depleted using 775 and 592 nm depletion lasers. Signals were acquisitioned using HyD SMD hybrid detectors (Leica) set at 658–698 nm for the ATTO 647N channel and 505–565 nm for the PicoGreen

channel. Images were acquisitioned using a 4× zoom factor and deconvolved using Huygens Professional with STED optical option (version 18.04, Scientific Volume Imaging). Images and cell diameter were analyzed using Fiji (Schindelin *et al*, 2012). Since cells displayed a variable morphology from ovoid to spherical, minor and major axes were measured and averaged to obtain a single representative cell diameter for each cell. Only cells exhibiting both signals were considered in the analysis.

### Transmission electron microscopy (TEM)

Exponential-phase *M. florum* cultures were centrifuged at 10°C for 15 min at $7,900 \times g$ and then washed three times with cold PBS1×. The supernatant was discarded, and cells were fixed at RT for 45 min and then overnight at 4°C by adding 1 ml of 2.5% $(w/v)$ glutaraldehyde on top of the cell pellet. Cells were then washed twice with PBS1×, post-fixed at RT for 90 min using a 1% $(w/v)$ osmium tetroxide solution, and washed twice with water. Cells were then dehydrated through a series of washes (5 min each) with 30, 50, 70, 85, 95%, and three times 100% $(v/v)$ ethanol. Samples were washed again three times using propylene oxide, with a 5-min incubation at RT after each wash. Samples were then incubated at RT for 1 h with 1:1 propylene oxide:Epon, incubated two times at RT for 180 min with pure Epon, and then overnight at RT with pure Epon. The Epon and cell mixture was embedded within a polyethylene capsule (BEEM) and polymerized by baking at 70°C for 48 h. The block was cut into thin sections (~ 80 nm) and placed on a copper grid, stained sequentially with uranyl acetate and lead citrate (~ 10 min each), and finally examined under a Hitachi H-7500 TEM microscope operating at an accelerating voltage of 80 kV. Images and cell diameter were analyzed using Fiji (Schindelin *et al*, 2012). Only cells with a clearly distinguishable cellular membrane, as shown in Fig 2A, were selected for diameter measurement. Since cells displayed a variable morphology from ovoid to spherical, minor and major axes were measured and averaged to obtain a single representative cell diameter for each cell.

### Measurement of buoyant cell density

*M. florum* buoyant cell density was assessed by discontinuous density gradient centrifugation in Percoll (GE Healthcare). First, a Stock Isotonic Percoll (SIP) solution was prepared by mixing nine parts $(v/v)$ of Percoll (GE Healthcare) to one part $(v/v)$ of 1.5 M NaCl, resulting into a 1.12 g/ml solution. The 100% $(v/v)$ SIP solution was then diluted with 0.15 M NaCl to obtain 80, 60, 40, and 20% $(v/v)$ SIP solutions, with corresponding densities of 1.10, 1.08, 1.05, and 1.03 g/ml, respectively. To easily differentiate density gradients, trypan blue was added to half of the dilutions (100, 60, and 20%) to a final concentration of 0.0008%. 2 ml of each dilution was then slowly layered from most concentrated to less into a 15-ml conical tube to create a discontinuous density gradient varying from 1.12 (100% SIP) to 1.03 g/ml (20% SIP). 20 ml of an exponential-phase *M. florum* culture was centrifuged at 10°C for 15 min at $7,900 \times g$ and then washed twice with cold PBS1×. Cells were resuspended in 2 ml of NaCl 0.15 M (1.00 g/ml) and slowly loaded on the top of the density gradient. Cells were then centrifuged two times at $7,900 \times g$ (10°C) for 30 min each, and the position of the cell pellet was noted after each centrifugation.

### Biomass quantification

Detailed biomass quantification methods are available in Appendix Supplementary Methods. A summary of the procedures is shown in Fig EV1. Briefly, dry mass was measured by weighting exponential-phase culture pellets previously dried at 80°C for ~ 36 h. Quantification was performed in quadruplicate and repeated three times. Protein mass was quantified in quadruplicate by fluorescence-based protein quantification of whole-cell lysates using the CBQCA Protein Quantitation Kit (Molecular Probes, C-6667) according to the manufacturer's specifications. DNA and RNA mass were quantified in quadruplicate by fluorescence-based nucleic acid quantification performed on purified genomic DNA and purified RNA using the Quant-iT PicoGreen dsDNA Assay Kit (Thermo Fisher Scientific, P7589) and the Quant-iT RiboGreen RNA Assay Kit (Thermo Fisher Scientific, R11490), respectively. Carbohydrate mass was measured in quadruplicate by GC-MS analysis performed on whole-cell lysates and normalized by the protein concentration of the samples. Lipid mass was quantified using a combination of two different methods: the identification of lipid species by direct infusion-tandem mass spectrometry (DI-MS/MS; see Dataset EV8) and the fluorescence-based quantification of phospholipids using the Phospholipid Assay Kit (Sigma-Aldrich, MAK122). Lipid quantifications were performed in quadruplicate. All quantifications were normalized by the cell concentration of analyzed samples using CFU or FCM counts.

### Protein mass spectrometry

The protein composition of *M. florum* was determined by 2D LC-MS/MS from trypsinized protein extracts. Sample preparation and analysis was executed by PhenoSwitch Bioscience (Sherbrooke, Canada). Briefly, an exponential-phase *M. florum* culture was centrifuged at 10°C for 2 min at $21,100 \times g$ and washed twice with cold electroporation buffer [272 mM sucrose, 1 mM HEPES (pH 7.4)]. Cells were then resuspended in 0.4% $(w/v)$ sodium deoxycholate and lysed using a Bioruptor UCD-200 sonication system (Diagenode) set at high intensity and 4°C for 35 cycles (30 s on, 30 s off). Insoluble material was removed by centrifuging the cell lysate at $16,000 \times g$ for 10 min at 4°C, and the supernatant was recovered. Protein concentration was measured using the Bio-Rad Protein Assay (Bio-Rad) according to the manufacturer's specifications and absorbance at 595 nm was measured using a Synergy HT microplate reader (BioTek). The cell lysate was then reduced at 65°C for 15 min with 10 mM dithiothreitol (DTT) in a final pH of 8.0 and then alkylated at RT in the dark for 30 min with 15 mM iodoacetamide. 10 mM of DTT was then added to quench residual iodoacetamide and proteins (~ 200 μg) were digested at 37°C overnight with shaking using 1 μg of trypsin per 30 μg of proteins. The resulting peptides were first separated using a polymeric reversed phase column (Phenomenex, 8E-S100-AGB) and eluted into eight fractions with increasing concentration of acetonitrile. ~ 5 μg of each fraction was then injected into a TripleTOF 5600 mass spectrometer (SCIEX) equipped with a HALO ES-C18 column (0.5 × 150 mm). Peptides were separated with a 60 min gradient of the following two mobile phases: (i) 0.2% $(v/v)$ formic acid and 3% $(v/v)$ DMSO in water; and (ii) 0.2% $(v/v)$ formic acid and 3% $(v/v)$ DMSO in ethanol. Peptides were analyzed in information dependant

acquisition (IDA) mode. Raw MS files were analyzed using Peptide-Shaker software version 1.13.4 (Vaudel *et al*, 2015) configured to run three different search engines (MS-GF+, Comet, and OMSSA) via SearchGUI (version 3.1.0) (Barsnes & Vaudel, 2018). SearchGUI parameters were set as follows: maximum precursor charge, 5; maximum number of post-translational modification per peptide, 4; precursor ion *m/z* tolerance, 0.006 Da; fragment ion *m/z* tolerance, 0.1 Da; maximum missed cleavages, 2; minimal peptide length, 8; and maximal peptide length, 30. Carbamidomethylation of C was set as a fixed modification. Acetylation of K, Acetylation of protein N-term, FormylMet of protein N-term, Oxidation of M, Phosphorylation of S, Phosphorylation of T, and Phosphorylation of Y were set as variable modifications. Protein search database was defined according to the published *M. florum* L1 RAST genome annotation (Baby *et al*, 2018). Peptide spectrum matches, peptides, and proteins were validated using a 1% FDR cut-off.

## Cell equations

For simplicity, we assumed *M. florum* cells to be of spherical shape in all cell equations described in this study since the observed morphology varied from ovoid to spherical (see Fig 2A). Given a spherical *M. florum* cell with a certain diameter (*d*), its volume (*V*), surface area (*A*), and surface area to volume ratio (*SA:V*) can be described according to the following equations:

$$V = \frac{\pi d^3}{6} \tag{1}$$

$$A = \pi d^2 \tag{2}$$

$$SA\text{:}V = \frac{A}{V} \tag{3}$$

Additionally, its cell mass (*CM*) can be described as follows:

$$CM = \frac{\pi d^3}{6} \times D \tag{4}$$

$$CM = \frac{DM}{DF} \tag{5}$$

$$CM = \left( \frac{\pi d^3}{6} - \frac{DM}{D_{DM}} \right) \times 1 + DM \tag{6}$$

$$CM = \left( \frac{\pi d^3}{6} - \frac{\frac{\pi d^3}{6} \times D \times DF}{D_{DM}} \right) \times 1 + DM \tag{7}$$

where *D*, *DM*, *DF*, and $D_{DM}$ are the cell buoyant density, dry mass, dry mass fraction, and dry mass-specific density. Detailed description of cell mass equations is given in Appendix Supplementary Methods. For each equation, the mean cell mass ($CM_{mean}$) was calculated using the mean value associated with each measured or estimated parameter. The minimal ($CM_{min}$) and maximal ($CM_{max}$) cell mass were calculated using the mean ± SD or the range associated with each parameter. For example, using Equation 4 and considering a cell buoyant density between 1.05 and 1.08 g/ml, the minimal and maximal cell mass values are given by the following expressions:

$$CM_{min} = \frac{\pi d^3}{6} \times 1.05 \tag{4.1}$$

$$CM_{max} = \frac{\pi d^3}{6} \times 1.08 \tag{4.2}$$

And the mean cell mass value is defined as follows:

$$CM_{mean} = \frac{\pi d^3}{6} \times 1.065 \tag{4.3}$$

According to typical ranges found in bacteria, the dry mass fraction (*DF*) and the dry mass specific density ($D_{DM}$) were estimated to be between 20–30% and 1.3–1.5 g/ml, respectively (Bakken & Olsen, 1983; Bratbak & Dundas, 1984; Bratbak, 1985; Fischer *et al*, 2009; Bionumbers, 2015). The most probable *M. florum* cell mass and cell diameter ranges were determined graphically according to the interception points of $CM_{mean}$ curves generated using a variable cell diameter in each equation (see Fig 2G). The most probable cell diameter range was finally used to infer the most probable cell volume (*V*) using Equation 1, as well as the most probable surface area (*A*) and surface area to volume ratio (*SA:V*) ranges using Equations 2 and 3.

## 5′-RACE library preparation and analysis

The 5′-RACE sequencing library was prepared from a *M. florum* exponential-phase culture as described previously (Carraro *et al*, 2014; Matteau & Rodrigue, 2015; Poulin-Laprade *et al*, 2015). Library quality and concentration were evaluated using a 2100 Bioanalyzer instrument (Agilent Technologies). Single-end Illumina sequencing (40 bp) was performed on an Illumina Genome Analyzer IIx instrument at the BioMicroCenter of the Massachusetts Institute of Technology (Cambridge, USA). Reads were trimmed for quality using Trimmomatic version 0.32 (Bolger *et al*, 2014) and aligned on *M. florum* L1 genome (NC_006055.1) with Bowtie 2 version 2.3.3.1 (Langmead & Salzberg, 2012). Alignments were processed and filtered to identify all putative TSSs. Analysis details are provided in Appendix Supplementary Methods. A summary of the 5′-RACE library statistics is shown in Appendix Table S1. Promoter motifs were searched using MEME and MAST version 5.0.3 (Bailey & Elkan, 1994). Strand-specific 1 bp resolution genome coverage tracks were generated using Bedtools genomecov version 2.27.1 (Quinlan & Hall, 2010).

## RNA-seq libraries preparation and analysis

Total RNA-seq libraries were prepared in biological triplicate from *M. florum* steady-state cultures grown using the Versatile Continuous Culture Device (Matteau *et al*, 2015). Total RNA was extracted in technical duplicate from each culture replicate using the Direct-zol RNA MiniPrep Kit (Zymo Research, R2052) as described previously (Carraro *et al*, 2014), for a total of six RNA-seq libraries. RNA-seq libraries were prepared and depleted from ribosomal RNA as described previously (Carraro *et al*, 2014), with the exception that 200 µg/ml of actinomycin D was added to the reverse transcription reaction to prevent second strand synthesis by the reverse transcriptase (Perocchi *et al*, 2007). Library quality and concentration were evaluated using a 2100 Bioanalyzer instrument (Agilent

Technologies). Paired-end Illumina sequencing (2 × 50 bp) was performed on a HiSeq 2000 Illumina instrument at the Plateau de biologie moléculaire et génomique fonctionnelle of the Institut de recherches cliniques de Montréal (Montréal, Québec, Canada). Reads were quality trimmed using Trimmomatic version 0.32 (Bolger et al, 2014) and aligned in a strand-specific manner on the M. florum L1 genome (NC_006055.1) with Bowtie 2 version 2.3.3.1 (Langmead & Salzberg, 2012). Reads with a MAPQ below 10 were discarded using samtools version 1.5 (Li et al, 2009). A summary of the RNA-seq library statistics is shown in Appendix Table S1. FPKM values were calculated for each M. florum L1 protein-coding gene (RAST annotation, see Baby et al, 2018) using the GenomicAlignments R package version 1.10.1 (Lawrence et al, 2013). Strand-specific 1 bp resolution genome coverage tracks were generated using Bedtools genomecov version 2.27.1 (Quinlan & Hall, 2010). Bedtools makewindows and multicov (version 2.27.1) were used to calculate the RNA-seq coverage on non-overlapping 1 kb windows for each replicate. Pearson's correlation coefficients between replicates (1 kb windows coverage as well as gene FPKM) were calculated using GraphPad Prism-integrated function (version 7.0a).

### Reconstruction of transcription units

Rho-independent terminators were predicted from M. florum L1 DNA sequence and genes annotation (RAST annotation, see Baby et al, 2018) using an updated version of the in-house Python script developed by de Hoon et al (2005). The main difference between the updated version and the original one is the replacement of the Mfold package (Mathews et al, 1999; Zuker, 2003) by the ViennaRNA package (Lorenz et al, 2011) (version 2.4.11) to calculate the RNA secondary structure. The Python script is available upon request from the author. Only terminators with a calculated score above 0 were considered significant. For each predicted terminator, the TTS was defined as the last base forming the stem-loop structure since the termination was shown to occur at or near the T-stretch following the stem-loop (Gusarov & Nudler, 1999; de Hoon et al, 2005). Strand-specific term-to-term scaffolds were then created according to the genomic position of the TTSs, and the coordinates of the motif-associated TSSs were used to generate all possible TUs for each scaffold. Genes were attributed to a given TU only if the calculated (5′-UTR) length was ≤ 500 bp and their coordinates were completely included within the TU, meaning that genes intersected with iTSSs were excluded from the iTSSs-derived TUs. Generated TUs were manually inspected using the UCSC genome browser (Kent et al, 2002) to correct for different scenarios such as the presence of predicted riboswitches (Kim et al, 2007; Kalvari et al, 2018), the circular topology of the chromosome or the occasional overlap between TSSs and terminator sequences. In the rare cases where no motif-associated TSSs could be attributed to a gene (orphan gene), the identified TSSs without promoter motif were considered for the expression of a TU encompassing that gene, as long as they initiated transcription on a purine nucleotide and fulfilled the other criteria described previously (signal intensity threshold and 5′-UTR length). If still no TSS without promoter motif could be found, then TSSs located at the end of the previous term-to-term scaffold (thus separated from the orphan gene by a predicted terminator) were considered as putative candidates for the expression of the gene, provided that its expression was non-null and the 5′-UTR length was

≤ 500 bp. See the manual curation notes column in Dataset EV3 for further details.

### Aggregate profiles

RNA-seq aggregate profiles were generated using the Versatile Aggregate Profiler (VAP) version 1.0.0 (Coulombe et al, 2014). Aggregate profiles were calculated for each DNA strand independently using the RNA-seq genome coverage calculated at single bp resolution on all the RNA-seq replicates merged together. The relative analysis method was used for all cases, along with two reference points and a 1 bp window size. The number of windows for the reference feature was set to 1 in the case of TSSs, whereas this parameter was set to 100 and 40 for TUs and terminators, respectively.

### Analysis of *Mesoplasma florum* ribosome-binding site

*Mesoplasma florum* ribosome binding site motif was determined by extracting the DNA sequence (20 bp) immediately upstream the translation initiation codon of each M. florum reference ORF and submitting it to MEME version 5.0.3 (Bailey & Elkan, 1994). The zero or one motif per sequence option was used, with a minimum motif length of 6 bp.

### Estimation of molecular abundances

The number of M. florum chromosome copies per cell was estimated from the measured DNA mass and the estimated molecular weight of the chromosome (see Table 1 for the measured DNA, RNA and protein). The molecular weight of the M. florum L1 chromosome (NC_006055.1) was estimated using the Sequence Manipulation Suite server (https://www.bioinformatics.org/sms2/dna_mw.html) (Stothard, 2000). The intracellular abundance of RNA species was calculated from the estimated molecular weight and measured RNA mass by assuming that rRNA, tRNA, and mRNA totalize 80, 15, and 5% of the total RNA mass of the cell (Westermann et al, 2012; Bionumbers, 2015). The molecular weight of RNA species was estimated using in-house Python scripts. The intracellular levels of protein species were calculated from the estimated molecular weight and the measured protein mass. The molecular weight of proteins was either estimated by PeptideShaker software version 1.13.4 (Vaudel et al, 2015) for proteins detected by mass spectrometry or using the Sequence Manipulation Suite server (https://www.bioinformatics.org/sms2/protein_mw.html) for proteins not detected by mass spectrometry (Stothard, 2000). For rRNAs and tRNAs, the total number of copies per cell was calculated by assuming that each species is found at equimolar ratios. For mRNAs and proteins, molar ratios were normalized according to the expression value of each species, i.e., using the associated FPKM and NSAF values, respectively. Briefly, the FPKM and NSAF values associated to each gene were divided by the sum obtained for all genes, resulting in a relative expression value. This value was then multiplied by the corresponding mRNA or protein molecular weight, producing a normalized molecular weight for each species, which was further divided by the sum of all normalized molecular weights to obtain a fraction of the total mRNA or protein mass for each gene. The mass of each mRNA and protein species was then calculated by multiplying mass fractions by the total mRNA and protein mass in M. florum, which was converted to an absolute number of molecules using their respective

molecular weight and the Avogadro number. Calculation details can be found in Datasets EV5 and EV6. The number of ribosomes per cell was estimated using two different approaches: (i) from the average number of protein per cell calculated for all predicted (KO) ribosomal proteins and (ii) by assuming that all rRNA molecules are incorporated into ribosomes, meaning that the estimated number of ribosomes per cell is equivalent to one third of the total number of rRNA molecules per cell (three rRNA molecules per ribosome). The number of RNAP complexes per cell was estimated according to the average number of protein per cell calculated for the α, β, and β′ subunits (see Dataset EV6). The protein stoichiometry of the RNAP complex was taken into account in the calculations (two α, one β, and one β′ subunits per RNAP).

### Analysis of functional categories expression

The KO Database was used to assign functional categories to *M. florum* reference proteins because of its clearly layered structure, and because major efforts were made to associate each KO entry with experimental evidences (Kanehisa *et al*, 2016a). Moreover, since proteins are assigned to functions via KO identifiers, the comparison between organisms is relatively straightforward. Briefly, the BlastKOALA server (https://www.kegg.jp/blastkoala/) (Kanehisa *et al*, 2016b) was used to assign KO identifiers to *M. florum* reference ORFs and retrieve associated functional categories. Since the same protein can be assigned to multiple functional categories, we then curated the assigned categories based on the non-redundant Proteomap functional hierarchy (Liebermeister *et al*, 2014). ORFs not matching to any KO identifiers were assigned to the unmapped category. KO entries matching to unclear functions were regrouped into the unclear category. Assigned KO identifiers and functional categories can be found in Dataset EV7. The Proteomap server (https://www.proteomaps.net/index.html) was used to visualize the relative expression of functional categories using either protein or mRNA expression datasets (Liebermeister *et al*, 2014).

### Data visualization

Raw 5′RACE and RNA-seq profiles, terminator and riboswitch predictions, identified TSSs, reconstructed TUs as well as identified peptide spectrum matches (PSMs) and validated peptides can be visualized using the UCSC genome browser at http://bioinfo.ccs.usherbrooke.ca/M_florum_hub.html.

## Data availability

The datasets produced in this study are available in the following databases:

- RNA-seq and 5′-RACE data: Gene Expression Omnibus GSE152985 (https://www.ncbi.nlm.nih.gov/geo/query/acc.cgi?acc=GSE152985)
- Proteomics data: PRIDE PXD019922 (https://www.ebi.ac.uk/pride/archive/projects/PXD019922)

**Expanded View** for this article is available online.

## Acknowledgements

We thank members of S. Rodrigue and P.-É. Jacques laboratories for helpful discussions, Joëlle Brodeur, Jean-Philippe Côté, Alain Lavigueur, Catherine Chamberland, and Pascal Sirand-Pugnet for their insightful comments on the manuscript, and Jean-François Lucier and the Centre de Calcul Scientifique at the Université de Sherbrooke for technical assistance. We are also grateful to Michiel J. L. de Hoon for sharing his Rho-independent terminators prediction script, as well as Charles Bertrand and the Plateforme de microscopie photonique at the Université de Sherbrooke for technical assistance on TEM and STED microscopy, respectively. Access to computational resources was provided in part by Calcul Québec (http://www.calculquebec.ca) and Compute Canada (http://www.computecanada.ca). This research project was funded by the Fonds de recherche du Québec—Nature et technologies: PR-173580, and by the Natural Sciences and Engineering Research Council of Canada: 386393.

## Author contributions

Manuscript writing and figure preparation: DM; Revision and editing of the manuscript and figures: SR, P-ÉJ, and J-CL; Experiments: DM, DG, SG, JMD and KD; Analyses: DM, FG, JMD, and J-CL; Project design: DM, J-CL, and SR; Preliminary data and insights on *M. florum*: TFK.

## Conflict of interest

The authors declare that they have no conflict of interest.

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
