## [Review Process File · Molecular Systems Biology]

Integrative characterization of the near-minimal bacterium *Mesoplasma florum*

Dominick Matteau, Jean-Christophe Lachance, Frédéric Grenier, Samuel Gauthier, James Daubenspeck, Kevin Dybvig, Daniel Garneau, Thomas Knight, Pierre-Étienne Jacques, and Sébastien Rodrigue

DOI: [10.15252/msb.20209844](https://doi.org/10.15252/msb.20209844)

Corresponding author(s): Sébastien Rodrigue (Sebastien.Rodrigue@USherbrooke.ca)

Review Timeline:

Submission Date:	6th Jul 20
Editorial Decision:	13th Aug 20
Revision Received:	13th Oct 20
Editorial Decision:	2nd Nov 20
Revision Received:	2nd Nov 20
Accepted:	3rd Nov 20

Editor: Jingyi Hou

Transaction Report:

Thank you for submitting your work to Molecular Systems Biology. We have now heard back from two of the three reviewers who agreed to evaluate your manuscript. Unfortunately, after a series of reminders we did not manage to obtain a report from reviewer #3. In the interest of time, and since the recommendations of the other two reviewers are quite similar, I prefer to make a decision now rather than further delaying the process.

As you will see from the reports below, the reviewers think that the study is interesting, and they acknowledge the potential relevance of the presented data. However, they raise a series of -mostly minor- concerns, which should be carefully addressed in a revision of the manuscript. Since the reviewers' recommendations are rather clear, there is no need to reiterate all the points listed below. Please feel free to contact me in case you would like to discuss in further detail any of the issues raised by the reviewers.

On a more editorial level, we would ask you to address the following issues.

REFEREE REPORTS

Reviewer #1:

I applaud the use of many complementary experiments and computational techniques to characterize the organism. This is an excellent study, and I have only a few minor and one major comments:

Major comment: When reading the results section, I frequently found myself wanting more discussion of the results and then the paragraph would end. I would have to go to the discussion to

obtain more information. This maybe a formatting requirement of the journal, but the Discussion might be more a summary of the major points. . For example, I was hoping that the reason for a smaller diameter in TEM measurements would be discussed when reading the results, but the reasons were not clear until the discussion. Given the large number of experiments integrated into characterizing the organism, it would have been more helpful if a brief discussion was presented along with the results.

Minor Comments:

Line 19: Count of genes in the organism should be made as early in the paper as possible. Especially as it is being compared to other organisms. How many of the genes have unknown or unclear assignments? What does not mapped mean in Table 2.

Line 111: The reader should have an idea of the total number of genes in *M. florum* before making this comparison, give number and genbank reference here. Also, state number of genes in JCVI-syn3.0 and JCVI-syn3A and any other organisms used in their comparisons and supplementary information.

Line 117: I would like some thoughts justifying why different genes between the two organisms are considered essential at this point. Gene essentiality can strongly depend on medium composition, so some discussion of that point should be included.

Line 206: I'm skeptical of these calculations. How much change in cell morphology might happen in STED / can you tell if the cells are still spherical in STED? If we trust more the diameters measured using STED. It would be good to see calculations based on them alone.

Line 314: This number (720 genes) should appear in the abstract or introduction.

Line 352: With approximately 2 copies of the chromosome in the cell, what would be the effects of DNA copy number on the mRNA and protein abundances?

Line 432: How do you arrive at the absolute number 250000 proteins. Details should be provided or briefly summarized.

Table 2S: Comparison of abundances in proteins and RNAs. On the column marked *Mycoplasma mycoides* you should put in parenthesis (Syn3A) as most of the values are from the elife 2019 article by Breuer et al. and the synthetic cell derived from *M. mycoides* is smaller than the parent organism.

Reviewer #2:

Summary

Mesoplasma florum is a very attractive model for defining the minimal set of genetic functions required for life. It has advantages over the related mycoplasma systems that are being used for this purpose because it is fast growing and appears to be easy to handle.

This study accumulates a massive amount of data characterizing the properties of *Me. florum*. This is nicely summarized in Figure 6 of the paper and includes a description of genome features, gene organization, molecular composition, physical properties, biomass composition, carbohydrate composition, and growth parameters. The value of such work lies in its potential usefulness in building computational models of the cell. This work is of high value in spite of its apparent lack of novelty. Its value lies in the degree of characterization, in one place, of cells grown under a set of reproducible conditions.

A broad range of methods was used, ranging from bioinformatic analysis of the genome sequence, to microscopic analysis, to detailed global analysis of cellular composition, to microbiological methods for characterization of growth properties. The value lies in having all this information collected using the same cells grown under identical conditions and reported in one place.

General remarks

Are you convinced of the key conclusions? Yes.
The work is carefully done and the results seem clear and reliable.

Place the work in its context

The data collected here will provide basic parameters needed in building a computational model of this attractive model organism.

What is the nature of the advance (conceptual, technical, clinical)?

The broad range of measurements gives this work its usefulness, not specific findings or concepts.

How significant is the advance compared to previous knowledge?

All of these measurements have been done on a the same near minimal organism. There is no comparable collection of data for any of the minimized mycoplasma genomes.

What audience will be interested in this study?

This will be of interest to those interested in defining the minimal genetic requirements for life.

Major points

Specific criticisms related to key conclusions. None.

The broad range of measurements gives this work its usefulness, not specific findings or concepts. One limitation of the work is that all the experiments are done with cells grown in rich medium. It would be very valuable to determine a defined medium that supports the growth of *Me. florum*. This would facilitate metabolic modeling of the cell.(Of course this isn't really a criticism of the work presented.)

Minor points

page 9, line156: "apparition" what is meant here? Perhaps "appearance"?

We would like to thank all reviewers for their thoughtful evaluation of our manuscript and highly constructive comments. We are very pleased that they showed a clear interest in our work and appreciated its value in the field. The comments raised by the reviewers were carefully addressed in this point-by-point response, and our manuscript was modified accordingly. The format of the manuscript was also verified and modified to address the issues raised by the editor to follow the Molecular Systems Biology guidelines. Important modifications include the transfer of Appendix Datasets S1-S8, Table S2, Figure S4, Figures S6-S8, and Figure S15 to the Expanded View (now referred to as Datasets EV1-EV8, Table EV1, Figure EV1, Figures EV2-EV4, and Figure EV5, respectively), as well as the reformation of the previous Materials and Methods Data Availability section into the formal format recommended by the journal. We are confident that the responses provided in this document will address concerns raised by the reviewers and that the modified version of the manuscript will be appropriate for publication in Molecular Systems Biology. Reviewer comments are in black while our responses are in blue.

Reviewer comments:

Reviewer #1:

I applaud the use of many complementary experiments and computational techniques to characterize the organism. This is an excellent study, and I have only a few minor and one major comments:

Major comment:

When reading the results section, I frequently found myself wanting more discussion of the results and then the paragraph would end. I would have to go to the discussion to obtain more information. This maybe a formatting requirement of the journal, but the Discussion might be more a summary of the major points. For example, I was hoping that the reason for a smaller diameter in TEM measurements would be discussed when reading the results, but the reasons were not clear until the discussion. Given the large number of experiments integrated into characterizing the organism, it would have been more helpful if a brief discussion was presented along with the results.

We agree that the Results section could have included more elements of discussion and detailed interpretations of the presented data, especially at the beginning of the section. However, considering the actual length of the manuscript and the amount of data and information already included in the Results section, we initially decided to limit discussion elements in this section to improve readability and instead regroup them in the Discussion. After revision of the manuscript, and by taking into account this comment made by Reviewer #1, we have modified a few sentences in the Discussion and Results sections to address this issue. These modifications aimed at reducing the number of minor points addressed in the Discussion. The following changes were applied:

Lines 159-160: the following sentence was added: “contrasting with pathogenic mycoplasmas such as *M. mycoides*, *M. capricolum*, and *Mycoplasma pneumoniae*.”

Lines 200-204: addition of TEM and STED cell diameter explanation previously described at lines 536-541 of the Discussion. This element was removed from the discussion.

Lines 241-242: the following sentence was added: “Overall, these results are comparable to fractions observed in other Mollicutes species (Razin *et al*, 1963)”. This sentence was removed from the discussion (previously at lines 562-564).

Lines 269-273: addition of the observation that the identified *M. florum* promoter motif is similar to promoter sequences identified in different Mollicutes, along with appropriate citations.

Lines 380-381: the number of chromosome copies per *M. florum* cell is now compared to estimations in JCVI-syn3A and *E. coli*. The reference to “2 copies” was also changed to “2.1 copies” to increase the precision of the approximation and for coherence with Table EV1 (see line 734 and Figure 6).

Line 423, lines 466-467, and lines 497-499: brief sentences comparing mRNA, protein, and ribosome concentrations between *M. florum* and other species (JCVI-syn3A, *M. pneumoniae*, *E. coli*) were added at the end of these paragraphs, removing the need of consulting the Discussion section to get the information.

Lines 616-660: this paragraph was reworded considerably to remove some details already well described in the Results section and to focus more on important points.

Minor Comments:

Line 19: Count of genes in the organism should be made as early in the paper as possible. Especially as it is being compared to other organisms. How many of the genes have unknown or unclear assignments? What does not mapped mean in Table 2.

We agree that this information should be available early in the paper. However, the exact number of genes varies among *M. florum* strains (see Baby *et al.* 2018; PMID: 29657968), and we would prefer to keep a more general statement in the abstract (and because the abstract is limited to 175 words). Instead, we have added an approximate *M. florum* genome size at line 35, which gives a better idea of how small the genome is, and provided the exact *M. florum* L1 gene count (along with GenBank accession number) at lines 108-120 of the introduction as well as in Table EV1.

In total, 272 protein coding genes (out of 685) have an unclear or unknown functional assignment based on the KEGG Orthology (KO) Database. 22 showed unclear assignments, while 250 could simply not be matched to any KO identifiers present in the database (not mapped). This information is available in Table 2, Dataset EV7, and at lines 472-477 of the manuscript. Methodology details are given at lines 1005-1109.

Line 111: The reader should have an idea of the total number of genes in *M. florum* before making this comparison, give number and genbank reference here. Also, state number of genes in JCVI-syn3.0 and JCVI-syn3A and any other organisms used in their comparisons and supplementary information.

Yes, good point. We added a sentence at lines 108-120 now clearly stating how many genes are predicted in *M. florum* L1 and in *M. mycoides capri* GM12, along with appropriate GenBank Accession numbers. We added a sentence at lines 111-112 for JCVI-syn3A, and specified the number of genes in JCVI-syn3.0 at line 106. We also modified the sentence at line 100 to indicate the JCVI-syn1.0 genome size. Genome size, GenBank accession number, and gene count were also added in Table EV1 for species selected in molecular abundance comparisons. The Table EV1 title was slightly modified to match with these changes.

Line 117: I would like some thoughts justifying why different genes between the two organisms are considered essential at this point. Gene essentiality can strongly depend on medium composition, so some discussion of that point should be included.

This is an excellent and exciting question that we are obviously very interested to investigate. However, many factors including medium composition, non-orthologous gene displacement as well as divergent evolutionary strategies could contribute to the differences observed in minimal genome compositions. This is now briefly mentioned in the Discussion at lines 778-781. While *M. florum* and JCVI-syn3A (*M. mycoides*) are routinely grown in two different complex and undefined rich medium (ATCC 1161 and

SP4) that share most if not all of the nutrients required for the growth of the two organisms, we would not expect to see a significant number of non-homologous essential genes explained strictly by differences in the culture medium. Relying on a semi or completely defined medium would surely help further exploring this question and investigating the potential cause of this phenomena. While this is beyond the scope of this publication, we are currently conducting high-density transposon mutagenesis in *M. florum* using a defined CMRL-based medium supplemented with different carbon sources that will help us address this question in an upcoming manuscript. Moreover, we will soon submit another manuscript describing a *M. florum* genome-scale metabolic model (GEM), which was used to perform minimal genome predictions that are next compared to JCVI-syn3.0 and JCVI-syn1.0.

Line 206: I'm skeptical of these calculations. How much change in cell morphology might happen in STED / can you tell if the cells are still spherical in STED? If we trust more the diameters measured using STED. It would be good to see calculations based on them alone.

TEM and STED have both their own biases and limitations, and we do not trust one method more than the other. As indicated in the manuscript (now at lines 200-204), cell dehydration in TEM often cause a reduction in cell diameter. In STED, the mounting media is known to have an impact on cell morphology, especially for bacteria and osmotically fragile species (see the following application note, which is now cited at line 190 of the manuscript:

<https://cdn.cytivalifesciences.com/dmm3bwsv3/AssetStream.aspx?mediaformatid=10061&destinationid=10016&assetid=27720>

This impact might have been exacerbated in our case since *M. florum* does not possess a cell wall, but we did not specifically investigate this aspect. A possible solution to prevent this problem would have been to use a soft mounting medium instead of a hardening mounting medium (Prolong Diamond in our case). Hardening media are known to compress samples in the z-axis as they cure and shrink. Alternatively, we could have tested different mounting media and evaluated their impact on *M. florum* morphology, e.g. by measuring their diameter in x, y, and z. However, while the x and y axes have a pretty good resolution (as low as 30 nm) with the STED microscope used in our study (Leica TCS SP8), the axial resolution has a resolution around 130 nm, and is known to be particularly affected by refractive index mismatches. Since it was not clear if we could get precise measurements of the z axis using our setup (and because performing z-stacks on multiple samples can be time consuming), we instead decided to use a mathematical approach that integrates other physical parameters to infer the total cell mass as well as refine range of cell diameters measured by TEM and STED. We think this approach provided results closer to the reality compared to only considering only the cell diameter obtained by STED alone. In fact, using the average STED cell diameter alone (741 nm) would mean that the *M. florum* cell has a total mass between 175 and 230 fg (see the curves in Figure 2G), which would be very surprising considering that we measured a total dry mass of 22.1 fg, thereby causing cells to have a dry matter content of only ~10% instead of ~20-30% as reported in other bacteria.

Line 314: This number (720 genes) should appear in the abstract or introduction.

Yes, this information was added at line 119 of the introduction and in Table EV1 (see previous points).

Line 352: With approximately 2 copies of the chromosome in the cell, what would be the effects of DNA copy number on the mRNA and protein abundances?

The is an interesting question. First of all, we have to keep in mind that our analyses were performed on a cell population and not on individual cells. The ~2 copies of the chromosome per cell therefore represents an average of the population; some cells probably carry 1 copy of the chromosome, while others could carry 2 or even 4 copies. Furthermore, this number represents only an “equivalent” of chromosome copies

based on the measured DNA mass per cell. The chromosome copies are not necessarily complete in actively replicating cells, and the replication machinery can initiate new rounds of replication on chromosome already being replicated. We can therefore expect genes near the chromosomal origin of replication (*oriC*) to have a higher copy number than those near the replication termination site (*ter*), which could result in higher expression both at the RNA and protein levels. We have briefly explored this scenario by looking at the relation between expression levels and the distance of any given gene from the *oriC* (see below). As previously reported by others (see Bryant *et al.* NAR 2014 and references therein), gene dosage seemed to have only very little impact on transcription levels, and no clear correlation could be observed. Virtually no effect was neither observed on protein expression levels. This suggests that other factors such as promoter strength, RNA stability, regulation mechanisms, and proximal genetic context likely have a more significant impact on gene expression than differences in gene copy numbers caused by chromosome replication.

Line 432: How do you arrive at the absolute number 250000 proteins. Details should be provided or briefly summarized.

The calculation details for absolute mRNA and protein numbers are now summarized at lines 1078-1086 of the Materials and Methods and are still available in Dataset EV5-EV6.

Table 2S: Comparison of abundances in proteins and RNAs. On the column marked *Mycoplasma mycoides* you should put in parenthesis (Syn3A) as most of the values are from the eLife 2019 article by Breuer *et al.* and the synthetic cell derived from *M. mycoides* is smaller than the parent organism.

Most of the abundance values were indeed obtained from Breuer *et al.* 2019 and represent estimates for JCVI-syn3A although the actual values were based on *M. mycoides capri* biomass data reported in Razin *et al.* 1963. Nonetheless, we now mention “(JCVI-syn3A)” in Table EV1 as suggested by the reviewer. To avoid any confusion, we also used the term JCVI-syn3A instead of *M. mycoides* subspecies *capri* at line 466 of the manuscript to compare protein concentrations. However, we have not seen any mention in the literature of JCVI-syn3A or JCVI-syn1.0 being smaller than the parent strain (*M. mycoides capri* GM12). In Gibson *et al.* 2010, it was reported that “both cell types (Syn1.0 and *mycoides*) show the same ovoid morphology and general appearance”, and in Hutchison *et al.* 2016 “Whereas syn1.0 grew in static culture as nonadherent planktonic suspensions of predominantly single cells with a diameter of ~400 nm”. If the reviewer is aware of a reference or report showing measurements of the *M. mycoides capri* cell diameter, we would happily include it in our manuscript, specifically at lines 574-575 and 744 of the Discussion. For now, we modified the sentences to also include syn1.0 and syn3A in the comparisons, presuming that their cell diameter is very similar to *M. mycoides capri*. We also added “(JCVI-syn3A)” at line 761 since the GEM was reconstructed for syn3A and not for *M. mycoides capri*, but the conclusions could also apply to *M. mycoides*. Finally, we replaced *M. mycoides capri* for JCVI-syn3A at line 763, and added the approximate doubling time of both organisms in this sentence (line 764).

Reviewer #2:

Summary:

Mesoplasma florum is a very attractive model for defining the minimal set of genetic functions required for life. It has advantages over the related mycoplasma systems that are being used for this purpose because it is fast growing and appears to be easy to handle.

This study accumulates a massive amount of data characterizing the properties of *Me. florum*. This is nicely summarized in Figure 6 of the paper and includes a description of genome features, gene organization, molecular composition, physical properties, biomass composition, carbohydrate composition, and growth parameters. The value of such work lies in its potential usefulness in building computational models of the cell. This work is of high value in spite of its apparent lack of novelty. Its value lies in the degree of characterization, in one place, of cells grown under a set of reproducible conditions.

A broad range of methods was used, ranging from bioinformatic analysis of the genome sequence, to microscopic analysis, to detailed global analysis of cellular composition, to microbiological methods for characterization of growth properties. The value lies in having all this information collected using the same cells grown under identical conditions and reported in one place.

General remarks:

Are you convinced of the key conclusions?

Yes. The work is carefully done and the results seem clear and reliable.

Place the work in its context

The data collected here will provide basic parameters needed in building a computational model of this attractive model organism.

What is the nature of the advance (conceptual, technical, clinical)?

The broad range of measurements gives this work its usefulness, not specific findings or concepts.

How significant is the advance compared to previous knowledge?

All of these measurements have been done on the same near minimal organism. There is no comparable collection of data for any of the minimized mycoplasma genomes.

What audience will be interested in this study?

This will be of interest to those interested in defining the minimal genetic requirements for life.

Major points:

Specific criticisms related to key conclusions.

None.

The broad range of measurements gives this work its usefulness, not specific findings or concepts. One limitation of the work is that all the experiments are done with cells grown in rich medium. It would be very valuable to determine a defined medium that supports the growth of *Me. florum*. This would facilitate metabolic modeling of the cell (of course this isn't really a criticism of the work presented).

We totally agree with the reviewer on this point and we are currently working on a defined CMRL-based medium for conducting future experiments such as high-density transposon mutagenesis.

Minor points:

page 9, line156: "apparition" what is meant here? Perhaps "appearance"?

“Apparition” was changed by the word “formation”.

Reviewer #3:

The manuscript "Integrative characterization of the near-minimal bacterium *Mesoplasma florum*" provides a high quality biophysical characterization and analysis of transcription of the mycoplasma *Mesoplasma florum*. As stated by the authors, the characterization provides "a strong experimental foundation for the development of a genome-scale model for *M. florum* and will guide future genome engineering endeavours in this simple organism". Both sets of data are exceptionally well done and analyzed. I cannot remember a better, more technically sound analysis of these aspects of any bacterium. Anyone that seeks to analyze a bacterium in anticipation of producing a whole cell computational model for that organism would be well served to use this paper as a guide for both biophysical measurements and transcriptome analysis. The authors are to be applauded for their work. Not only are the experiments done just the ones I think should have been done, the text explaining the findings was clean and clear. I do propose a small set of very minor changes/issues listed below that I think might make the paper a little clearer.

Line 19 Change "constitutes" to "is"

This was modified accordingly.

Lines 149-151 The exponential phase coincided with an important decrease in OD560nm easily noticeable by a growth medium color change from red to orange, corresponding to a drop in medium pH (from ~8.0 to 6.5).

This sentence is unclear. It might be interpreted that the decrease in OD560nm is easily noticeable rather than the pH change causing the phenol red in the media to change color.

The sentence was reformulated as follows to avoid misinterpretation (lines 168-171): “The exponential phase coincided with a substantial drop in medium pH (from ~8.0 to 6.5) causing the phenol red present in the culture medium to change color from red to orange, corresponding to an important decrease in measured OD_{560nm}.”

Lines 152-154 CFU and FCM cell concentrations were highly consistent with each other until late stationary phase, where they culminated at ~1x10¹⁰ cells/ml.

I suggest saying the CFU and FCM concentration measurements diverged at ~1x10¹⁰ cells/ml. Culminated is not the correct word here.

The sentence was modified as follows (lines 174-175): “CFU and FCM cell concentrations were highly consistent with each other until late stationary phase, where they both reached a plateau at ~1x10¹⁰ cells/ml and started to diverge.”

Line 156 This was followed by a gradual apparition of cell aggregates in the culture,

Rather than apparition, say appearance.

“Apparition” was changed by the word “formation”.

Line 211 Two thirds not two third

This was modified accordingly.

Line 215 The majority of the carbohydrate fraction most probably account for....

This should be accounts, not account

This was modified accordingly.

Line 472 Change "constitutes" to "is"

This was modified accordingly.

Line 742 This should probably be JCVI-syn3A, which was the subject of the Breuer *et al.* 2019 computational modeling study.

Yes, good observation. Other mentions of *M. mycoides*, JCVI-syn1.0, syn3.0, and syn3A were also verified and corrected when necessary (view previous points).

Lines 954-956 RNA-seq libraries were prepared and depleted from ribosomal RNA as described previously (Carraro et al, 2014), with the exception that 200 $\mu\text{g/ml}$ of actinomycin D was added to the reverse transcription reaction.

A reference is needed to explain the use of actinomycin D.

A brief explanation for the use of actinomycin D is now given at line 1003-1004, and the appropriate reference was added at the end of the sentence (Perocchi *et al*, 2007).

2nd Editorial Decision

2nd Nov 2020

Thank you for sending us your revised manuscript. We are now satisfied with the revision and I am pleased to inform you that your manuscript will be accepted in principle pending the following essential amendments.

2nd Authors' Response to Reviewers

2nd Nov 2020

The Authors have made the requested editorial changes.

Accepted

3rd Nov 2020

Thank you again for sending us your revised manuscript. We are now satisfied with the modifications made and I am pleased to inform you that your paper has been accepted for publication.

Corresponding Author Name: Sébastien Rodrigue
Journal Submitted to: Molecular Systems Biology
Manuscript Number: MSB-20-9844